# GROKKING IN PRETRAINING? MONITOR LLM'S MEMORIZATION-TO-GENERALIZATION WITHOUT TEST

**Ziyue Li, Chenrui Fan**
Department of Computer Science, University of Maryland, College Park
`litzy619@umd.edu, cfan42@umd.edu,`

**Tianyi Zhou**
MBZUAI
`tianyi.zhou@mbzuai.ac.ae`

## ABSTRACT

This paper presents *the first study of grokking in practical LLM pretraining*. Specifically, we investigate when an LLM memorizes the training data, when its generalization on downstream tasks starts to improve, and what happens if there is a lag between the two. Unlike existing works studying when a small model generalizes to limited and specified tasks during thousands epochs' training on algorithmic data, we focus on a practical setting for LLMs, i.e., near single-pass pretraining of next-token prediction on a cross-domain, large-scale corpus, and generalization on diverse benchmark tasks covering math/commonsense reasoning, code generation, and domain-specific retrieval. Our study, *for the first time, verifies that grokking still emerges in pretraining mixture-of-experts (MoE) LLMs*, though different local data groups may enter their grokking stages asynchronously due to the heterogeneity of their distributions and attributions to others. To find a mechanistic interpretation of this local grokking, we investigate the dynamics of training data's pathways (i.e., expert choices across layers in MoE). Our primary discovery is that *the pathways evolve from random, non-smooth across layers, instance-specific to more structured and transferable across samples*, despite the converged pretraining loss. This depicts a transition from memorization to generalization. Two novel metrics are developed to quantify these patterns: one computes the pathway similarity between samples, while the other measures the consistency of aggregated experts between subsequent layers for each sample. These training data based metrics induce near zero cost but can faithfully track and monitor the generalization of LLMs on downstream tasks, reducing reliance on costly instruction tuning and benchmark evaluations. We also ground our findings in a theoretical analysis of one-layer MoE, showing that more structured pathways improve the generalization bound.

## 1 INTRODUCTION

Large language models (LLMs) with Transformer (Vaswani et al., 2017) architectures have demonstrated that pretraining for a general-purpose task such as autoregressive prediction on a large corpus can gain powerful generalization across diverse downstream tasks and domains (Brown et al., 2020; Touvron et al., 2023; Bai et al., 2023; Zhao et al., 2025). Capabilities such as prompting (Wei et al., 2023), reasoning (Huang & Chang, 2023; Ahn et al., 2024), and in-context learning also emerge during training. However, a counter-intuitive "grokking" (Power et al., 2022; Nanda et al., 2023; Tan & Huang, 2024; Lv et al., 2025) has been widely observed in training Transformer models: the generalization starts to improve or even sharply rises long after training loss converged or plateaued—an indicator of overfitting in conventional machine learning. While grokking indicates a transition from memorization to generalization (Huang et al., 2024), it challenges our fundamental understanding of training dynamics and the underlying mechanism of generalization.

Despite recent attempts to explain the generalization induced by grokking, they have been largely confined to small models trained for hundreds to thousands of epochs on algorithm-synthesized small-scale data with generalization to limited and highly specified tasks (Power et al., 2022; Liu

et al., 2022a; Nanda et al., 2023; Merrill et al., 2023; Humayun et al., 2024; DeMoss et al., 2024; Prieto et al., 2025). Instead, LLMs' pre-training often takes only one epoch on a web-scale corpus of heterogeneous, cross-domain data that usually contains noise, imbalance, and redundancy (Brown et al., 2020; Zhao et al., 2025). Hence, LLMs' learning dynamics can vary drastically across different data, leaving whether grokking still emerges in LLM pretraining an open problem. It is also unclear whether and when different data are memorized with converged loss, and, whether they are memorized at the same time. Moreover, without the repeated replay of the same data and a decrease in loss, the conversion from memorizing training tokens to generalizing on downstream tasks remains mysterious: When does the model start to generalize to different tasks? Is memorization a prerequisite? How does generalization performance change after memorization?

Unfortunately, these questions cannot be answered by the dynamics of pretraining loss since the generalization performance can still vary after the loss plateaued, as observed in previous studies of grokking (Power et al., 2022; Nanda et al., 2023). For LLMs, it is also practically expensive to monitor its generalization performance during pretraining, since the model has to be finetuned to gain instruction-following capability before being evaluated on downstream tasks. Hence, developing an efficient-to-compute metric to accurately track the generalization can offer critical insights on better understanding and improving the pretraining of LLMs.

**Main Contributions** In this paper, we investigate the training dynamics and grokking in LLM pretraining for a mechanistic interpretation of its emergent generalization on downstream tasks. Our study is conducted on the open-sourced pretraining checkpoints of a 7B mixture-of-experts (MoE) LLM, OLMoE (Muennighoff et al., 2024). While our study mainly focuses on OLMoE, its pretraining and evaluation setups are common among other LLMs. For the first time, we verify the existence of local grokking during LLM pretraining. However, unlike the previously observed global and synchronous grokking for most data (Power et al., 2022; Merrill et al., 2023; Nanda et al., 2023), *LLM memorizes domains/groups of training data at different training steps and takes varying amounts of steps to generalize to related benchmark tasks.* The starting time and lasting steps of such local grokking vary across data groups. As a result, generalization performance is usually unstable during the earlier pretraining stages but starts to steadily improve once sufficient data have been memorized. This local difference also reflects data difficulty: the later memorized data group often takes longer to generalize.

Another primary challenge is to explain how the continual pretraining after memorization completed leads to the delayed sharp rise of the generalization performance. To investigate the mechanism of memorization-to-generalization transition, we explore internal states of LLMs track their major changes after loss converges. Specifically, we study how the pathways (the expert choices across layers for training samples) in OLMoE change. We introduce two metrics to measure their complexity how it changes over time: (1) the edit distance between different samples' pathways; and (2) the consistency of selected experts' embedding between consecutive layers for each sample.

Our empirical analysis on four domains reveals a prominent trend of the pathway complexity during grokking: the edit distance declines and the consistency increases, despite the converged training loss and more data being learned. This change in pathway complexity reflects a transition to smarter memorization: it keeps discovering more transferable knowledge across samples to encode data more efficiently. *It explains why generalization improves with plateaued training loss: the model keeps finding more generalizable and shareable structures to memorize training data.* Additionally, we provide a theoretical explanation relating the pathway complexity to a generalization bound of a one-layer MoE. Practically, the two metrics show strong correlations with the evaluation results on standard benchmarks after instruction finetuning. Therefore, they provide a zero-cost, finetuning and benchmark evaluation-free tool to monitor generalization during pretraining, which is critical to LLM developers and practitioners. These discoveries take the first step toward bridging the gaps of understanding grokking and memorization-to-generalization transition in LLM pretraining. Our key findings and contributions can be summarized as:

- Grokking still occurs during the near single-pass pretraining of practical-scale LLMs, but it is local and asynchronous for different data groups, unlike global grokking for all data in previous works.

- Grokking's memorization-to-generalization transition can be explained by the dynamics of pathways in MoE: pathway similarity between samples and consistency across layers increase. A theoretical connection is also built between the pathway complexity and a generalization bound.

- The two metrics we developed to measure pathway complexity provide a finetuning and evaluation-free tool with almost zero cost to monitor the generalization during LLM pretraining.

## 2    RELATED WORK

Previous studies on grokking have primarily focused on relatively small models and simplistic tasks. Grokking was first identified by Power et al. (2022), who observed the phenomenon using a two-layer transformer trained on a small-scale algorithmic dataset. Subsequent investigations have extended these findings to shallow transformers and densely connected networks (Nanda et al., 2023; Notsawo Jr et al., 2023; Merrill et al., 2023; Liu et al., 2022b; Fan et al., 2024), and CNNs (Humayun et al., 2024). Notably, all aforementioned studies employ multi-epoch training paradigms, which differ substantially from the one-pass training approach typical in LLM pretraining. Although Lv et al. (2025) examined grokking behavior using a 162M parameter transformer pretrained on 40 billion tokens for a copying task, the scale of their model, dataset, and the simplicity of the task remain distant from practical, real-world scenarios. In contrast, our study is the first to investigate grokking in a 7-billion parameter LLM across diverse and practical tasks, including mathematical reasoning, code generation, commonsense understanding, and domain-specific knowledge retrieval.

Several works have focused on revealing the mechanisms behind grokking behavior. Liu et al. (2022a) demonstrated in a simplified setting that grokking results from structured representations emerging during training. Nanda et al. (2023) further revealed that grokking can be attributed to the gradual amplification of structured mechanisms encoded within model weights. Wang et al. (2024) showed that even small Transformers trained from scratch on synthetic reasoning tasks exhibit a grokking-driven emergence of implicit reasoning, where generalization arises only after extended overfitting and internal structure formation. Using a single-layer ReLU network, Merrill et al. (2023) associated grokking with subnetwork sparsity. Meanwhile, Humayun et al. (2024) observed that decision boundaries become smoother during the grokking phase in CNN classification tasks. Beyond interpretability-focused research, Notsawo Jr et al. (2023) identified oscillatory patterns in early training loss as predictors of subsequent grokking phenomena. Distinct from these works, our research provides an explanation of grokking within significantly larger LLMs through an interpretative analysis of MoE architectures. Additionally, we introduce practical indicators for monitoring generative behavior throughout LLM pretraining.

## 3    GROKKING (DELAYED GENERALIZATION) IN LLM PRETRAINING

While most existing works study grokking of small models on specified algorithmic tasks, investigating similar phenomena in LLM pretraining poses several new unique challenges, which inherently motivate several designs in our experimental settings.

- **Training-test evaluation gap**: most LLMs are trained for the next token(s) prediction while evaluated on diverse benchmark tasks, making the conventional comparison between training and test loss/accuracy invalid. Hence, we need to adapt the definition of grokking to the LLM setting.

- **Heterogeneous, web-sourced, cross-domain training data**: While previous grokking studies focus on clean training data with the same distribution/format, most LLMs are pretrained on heterogeneous, cross-domain, web-sourced data that contain redundant, contaminated, or imbalanced samples. Hence, their loss may converge at different rates and pose distinct effects on downstream tasks. To address these challenges, we carefully filter and group the data in our analysis.

- **Generalization to diverse downstream tasks**: Previous studies of grokking focus on generalization to highly-specified and limited tasks, while practical LLMs are expected to generalize to arbitrary tasks or domains via prompting. Hence, LLMs' generalization behaviors may vary drastically across different downstream tasks/benchmarks and difficulty levels during pretraining.

- **Near single-pass pretraining**: Most LLMs' pretraining only takes approximately one epoch, so the model has been trained only once on each sample, and its training loss may even converge before visiting later data. However, such "memorization" of unseen data is not caused by overfitting or repeated replay. Instead, it reflects the interpolation or composition of learned data, which might be counted as "generalization" in previous works but differs from the emergent generalization of LLMs on new tasks.

## 3.1 STUDY TRAINING DYNAMICS OF LLM PRETRAINING

**Memorization is measured by the pretraining objective.** Next token prediction (NTP) is the most widely adopted pretraining objective for LLMs. Let the pretraining corpus be $D_{\text{train}}$, it aims to minimize the negative log-likelihood (NLL) of each token $x_{i,j}$ given previous tokens $x_{i,<j}$ in a text sequence $x_i = (x_{i,1}, \ldots, x_{i,|j|})$ drawn from $D_{\text{train}}$:

$$\ell_i(\theta) = -\frac{1}{|x_i|} \sum_{j=1}^{|x_i|} \log p_\theta(x_{i,j} \mid x_{i,<j}).  \tag{1}$$

Since $\ell_i(\theta)$ directly measures how the model memorizes each token in $x_i$, we will track it in the pretraining process and identify the sample with consistently small and converged loss across checkpoints as memorized. Specifically, for a sample $x_i$ with loss $\ell_i(\theta_t)$ at pretraining step $t$, we identify it as *memorized* since step $t_i^*$ if $\ell_i(\theta_t) \leq \varepsilon$ and $|\ell_i(\theta_t) - \ell_i(\theta_T)| \leq \delta$ for all $t \geq t_i^*$, where $\varepsilon$ is a small threshold, $\delta$ enforces stability over time, and $T$ indexes the final pretraining checkpoint. In Appendix E.3, We verify that varying $\varepsilon$ and $\delta$ does not change the post-memorization behavior analyzed in Section 4.

**Generalization is measured on standard benchmarks after instruction tuning.** As foundation models with various emergent capabilities, LLMs are expected to generalize to an open set of unseen tasks through simple prompting/instructions. Hence, their generalization performance is usually measured across standard benchmarks defined on diverse downstream tasks. To equip a pretrained LLM $\theta$ with the instruction-following capability, an instruction tuning algorithm $\mathcal{A}_{\text{IT}}$ further finetunes $\theta$ on a dataset $D_{\text{IT}}$. The generalization performance on benchmark-$i$ is then measured by

$$\mathcal{G}_i = \text{Eval}_i(\theta_{IT}, D_i), \theta_{IT} = \mathcal{A}_{\text{IT}}(\theta, D_{\text{IT}}),  \tag{2}$$

where $D_i$ is the dataset in benchmark-$i$ and $\text{Eval}_i$ defines the corresponding evaluation metric, which varies across benchmarks for different types of tasks. For example, accuracy is usually used in multi-choice QA tasks while BLEU/ROUGE scores are used in summarization and generation tasks. For code generation tasks, the generalization performance is usually measured by unit tests, where a test case is considered correct if the generated code can pass the unit tests.

**Grokking refers to delayed generalization improvement after memorization is completed.** Following the same spirit of previous works on grokking, we mainly focus on the phenomenon that $\mathcal{G}_i$ starts to surge long after $\ell_i(\theta)$ has converged/plateaued. We also use the training loss to track memorization as previous works. However, the test loss adopted by previous grokking analysis is no longer an ideal metric for LLM generalization, since it focuses on token-level matching to ground truths while downstream tasks focus on semantic-level matching, and some of them do not provide ground truths (e.g., code generation).

**Experimental Setup** We equidistantly sample 10 checkpoints from the OLMoE pretraining trajectory. To evaluate memorization, we probe OLMoE on samples drawn directly from its pretraining corpus $D_{\text{train}}$, ensuring fidelity to the model's original distribution. We evaluate generalization on widely used benchmarks spanning four domains: *math*, *code*, *commonsense QA*, and *domain-specific QA* (10k pretraining and 10k test samples per domain; see Appendix A). To exclude contaminated benchmark samples from the generalization analysis, we apply the Min-K%++ membership inference method (Zhang et al., 2024) to identify them. Moreover, we mainly focus on the samples whose model predictions become consistently correct before the pretraining ends, and analyze when this generalization emerges and how long it persists. For downstream tasks, we apply lightweight LoRA-based instruction tuning $\mathcal{A}_{\text{IT}}$ on $D_{\text{IT}}$ to equip checkpoints with basic instruction-following capability while preserving pretrained capabilities (Appendix B).

## 3.2 ASYNCHRONOUS MEMORIZATION, DELAYED GENERALIZATION, & LOCAL GROKKING

To understand when memorization unfolds during pretraining and how it affects generalization, Figure 1 contrasts the number of memorized training samples (using the criterion defined above) at each pretraining checkpoint with benchmark performance for each domain. It shows that (1) training samples are memorized at different pretraining steps; (2) generalization typically follows memorization with a lag: benchmark performance starts to achieve stable gains after sufficient data has been memorized. An interesting discovery is that the lag length varies across domains: generalization

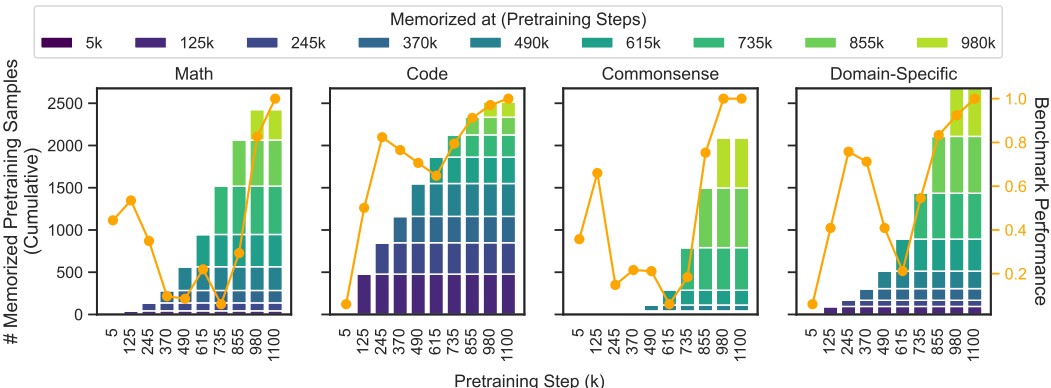

Figure 1: **Memorized training samples at different pretraining steps and the generalization performance** on standard benchmarks for each domain. It shows asynchronous memorization of different data in each domain and a delayed generalization leap after memorizing a certain amount of data.

leap on math and coding tasks requires memorizing many more samples than commonsense and domain-specific QA tasks.

These observations reveal the complex training dynamics of LLMs caused by training data heterogeneity and their uneven attributions to different benchmark tasks. It implies that the widely studied global grokking—a synchronous memorization of most training data and a delayed contemporary generalization leap on most test samples—cannot be found in LLM pretraining. That being said, the memorization of certain data is plausible to impose a long-term, delayed effect on the generalization to specific downstream tasks. To investigate such local dynamic patterns, we group data in each domain by their memorization steps $t_i^*$ and particularly focus on the early-memorized groups, as they leave more checkpoints after $t_i^*$ to study the subsequent dynamics of generalization. Accordingly, we group samples in benchmarks by the training steps since when their predictions become consistently correct. To enable an analysis of how memorization drives downstream generalization performance, we pair each training data group with its most relevant benchmark samples via bipartite graph matching (by Hungarian algorithm (Kuhn, 1955)), with the training-test data similarity defined in the embedding space of SentenceTransformer (Reimers & Gurevych, 2019). Additional implementation details of the pairing procedure are provided in Appendix B.1.

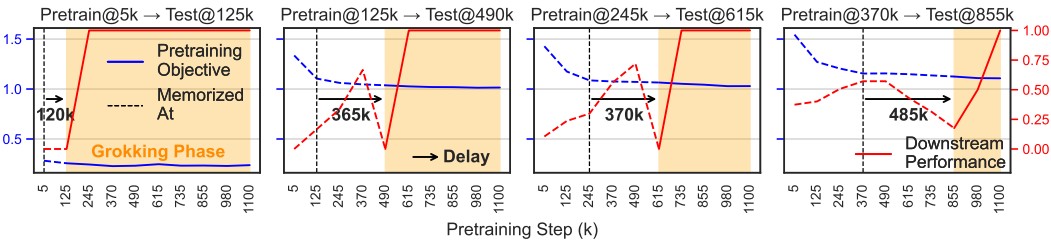

Figure 2: **Pretraining objective (blue) and benchmark performance (red) across four paired training-test data groups from the math domain.** Pretrain@$p \to$ Test@$q$ denotes a paired training–test group, where Pretrain@$p$ is the training subset whose samples reach stable memorization at step $p$, and Test@$q$ is the benchmark subset whose accuracy stabilizes at step $q$. These pairs are formed using embedding-based Hungarian matching, as detailed in Appendix B.1. We observe local grokking on each pair: a delayed generalization leap on benchmark tasks after the pretraining objective plateaued. From left to right, more difficult data memorized (converged) later associate with a longer delay towards grokking.

Figure 2 visualizes the memorization and generalization on four groups: all training-test pairs exhibit a delayed generalization leap on benchmark tasks after the pretraining objective stably converged on the training data. Hence, the training objective cannot monitor or predict the generalization performance. Moreover, earlier-converging groups generalize sooner after loss converged, while later-converged groups associate with substantially longer delays. Notably, this delay scales with data difficulty, which affects not only the number of steps taken to memorize the data but also the duration of memorization to generalization transition.

**Remaining Question** This local grokking indicates that the model's internal states, except its pretraining loss, might track generalization or reflect the transition better. This raises a fundamental question for understanding the emergence of capability in LLM: *What internal state changes within an LLM signify the emergence of generalization?* Investigating this problem can provide critical insights to uncover the mechanisms of memorization-to-generalization transition and monitor the pretraining progress in practice. To this end, we probe LLMs' internal state dynamics to address the challenge.

## 4 MONITOR MEMORIZATION-TO-GENERALIZATION TRANSITION BY TRAINING DYNAMICS OF ROUTING PATHWAYS

As shown in Section 3, generalization in LLMs emerges well after memorization, suggesting that internal reorganization underlies this transition. Such dynamics are difficult to track in dense architectures, where all neurons are simultaneously involved. MoEs simplify this process: by organizing computation into experts, each input activates only a subset of experts, forming a discrete *routing pathway* across layers (Figure 3(a)). These pathways reveal how computation is allocated and reorganized during pretraining, providing a mechanistic view of the memorization-to-generalization transition.

In this section, we analyze the evolution of *routing pathway* during pretraining and introduce two metrics to quantify these changes (Figure 3): (i) the similarity of pathways across different inputs (Section 4.1), and (ii) the consistency of a single input's pathway (Section 4.2). Importantly, all metrics are computed on pretraining samples that have already been memorized, i.e., those whose training objective has stabilized at low values beyond specific checkpoints, and trace how their internal routing continue to evolve from memorization to generalization. We then assess these metrics as robust generalization monitors (Section 4.3) and conclude by establishing a spectral framework connecting routing geometry to generalization bounds (Section 4.4).

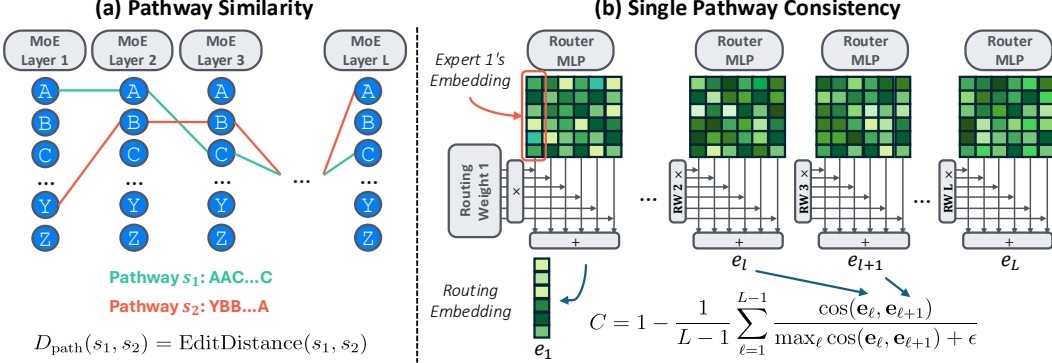

Figure 3: **Pathway Complexity Metrics to Monitor Grokking.** (a) Pathway similarity between samples is measured by edit distance on their sequences of expert choices across layers. (b) Pathway consistency quantifies the smoothness of expert transitions between subsequent layers for the same sample by cosine similarity of their weighted expert embeddings.

### 4.1 PATHWAY DISTANCE AND KNOWLEDGE SHARING BETWEEN TRAINING SAMPLES

Building on prior findings that grokking coincides with the emergence of *structured internal mechanisms* (Liu et al., 2022a; Nanda et al., 2023; Wang et al., 2024), we hypothesize that as the model begins to generalize, it transitions from highly individualized, input-specific computation toward more *shared, structured computation across related inputs*. This shift should manifest as increasing **pathway similarity**—the convergence of routing patterns across similar samples—indicating that the model is developing common mechanisms for solving related tasks.

We test this by examining how the similarity of expert pathway evolves during pretraining. Each input's pathway is the ordered sequence of selected experts through the $L$ MoE layers. Specifically, for input $x_i$, we define its pathway $s_i$ as: $s_i = \text{concat}(e_1^{(i)}, e_2^{(i)}, \ldots, e_L^{(i)})$, where $e_\ell^{(i)}$ denotes the ordered list of expert indices at layer $\ell$, obtained by ranking experts according to the mean routing weights across all tokens for input $x_i$. To determine which experts are included, we first rank experts based on their routing weight and then select top-ranked experts until their cumulative routing weight

exceeds a predefined threshold. The weight threshold selects the experts that actually drive each input's computation, rather than picking a fixed number of experts. In our main analysis, we use a cumulative routing-weight threshold of 0.7, and Appendix E.1 shows that the observed pathway trends remain consistent when varying this threshold. The selected experts are then concatenated as a comma-separated string (e.g., '3,1,5') which are later joined across layers with hyphens (e.g., '3,1,5-...-9,1').

We use the Levenshtein edit distance $D_{\text{path}}(s_i, s_j) = \text{EditDistance}(s_i, s_j)$, which counts the minimum insertions, deletions, or substitutions required to align two pathways, to measure pathway similarity. This metric captures both local deviations in expert choice and global shifts in pathway structure, reflecting differences in selection, length, and ordering. We monitor the average pairwise distance across samples throughout pretraining to track the emergence of shared routing.

**Overall pattern.** Our analysis, shown in Figure 4, reveals a clear trajectory in pathway dynamics. Early in pretraining, most inputs share nearly identical routes, producing low edit distance $D_{\text{path}}$. As the model memorizes individual inputs, pathways diverge and $D_{\text{path}}$ rises. Crucially, $D_{\text{path}}$ decreases after memorization, which indicates that semantically related inputs then converge into similar routing sequences, signaling the **emergence of shared pathways** that support generalization.

**Layerwise pattern.** While prior work (Lo et al., 2025) reports static depth-dependent specialization in trained MoEs, our results show that the memorization-to-generalization transition is also dynamically depth-dependent during pretraining (Figure 5). Shallow layers show the steepest decline in $D_{\text{path}}$ after memorization, indicating rapid convergence to shared pathways. Deeper layers exhibit smaller reductions, and the final layer even shows an increase in $D_{\text{path}}$ post-memorization. These patterns suggest a depth-specific reorganization: early layers consolidate universal representations, while later layers retain or enhance routing flexibility, balancing shared computation with task-specific specialization.

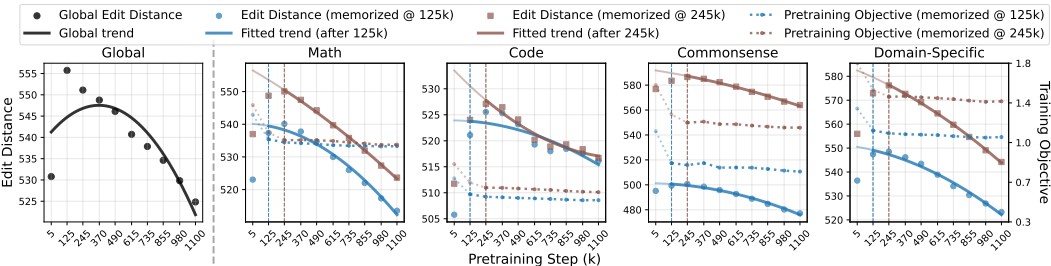

Figure 4: **Pathway edit distance and pretraining objective of two data groups** (memorized at $125k$ and $245k$ steps) during pretraining across four domains. The global panel (leftmost) fits the overall quadratic trend over all domains, while domain panels fit separate trends after the corresponding convergence step. Despite early training objective plateauing, pathway edit distance continues to decline, indicating a declining complexity of internal memorization.

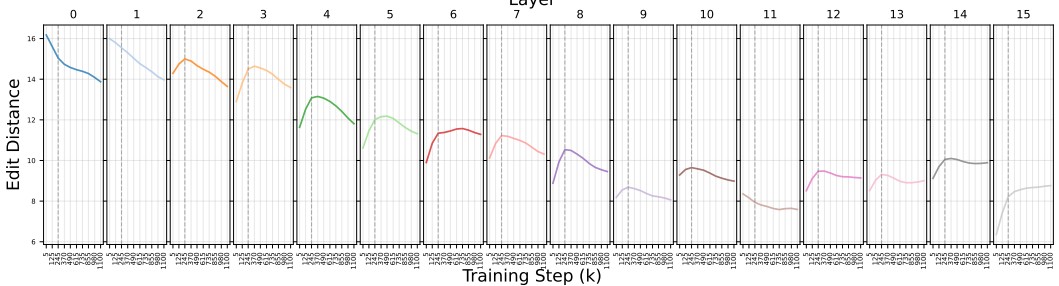

Figure 5: **Layer-wise evolution of pathway edit distance during pretraining** for the data group memorized at $245k$ step. Overall distance decreases from early to later layers, indicating higher complexity in early layers. After pretraining objective convergence, early-layer routing among similar samples simplifies quickly, later layers adapt more gradually, and the final layer diversifies after memorization.

## 4.2 PATHWAY CONSISTENCY AND COMPLEXITY FOR SINGLE SAMPLES

We next analyze how the routing complexity of individual inputs evolves during pretraining. Our hypothesis is that increasingly streamlined and consistent expert selection across layers reflects the reuse of shared pathways, signaling the emergence of generalizable processing strategies. To capture this, we define **single-sample pathway consistency**, which measures the smoothness of expert transitions across consecutive layers.

In a typical MoE, the router at layer $\ell$ is a single-layer MLP—a linear transformation followed by a softmax—producing scores over $K$ experts, with weight matrix $\mathbf{W}_\ell \in \mathbb{R}^{K \times d}$. We define the *expert embedding* for expert $k$ as $\mathbf{v}_\ell^k := \mathbf{W}_\ell[k], k = 1, \ldots, K$, which serves as an anchor vector to compute routing scores. For input $x_i$, its *routing embedding* is the weighted sum:

$$\mathbf{e}_\ell^{(i)} = \sum_{k=1}^K g_\ell^{(i,k)} \cdot \mathbf{v}_\ell^k,$$

where $g_\ell^{(i,k)}$ is the routing weight assigned to expert $k$. Although experts are independently parameterized, the strong correlations between adjacent layers in deep networks (He et al., 2016) and MoEs (Li & Zhou, 2025) suggest that $\{\mathbf{v}_\ell^k\}$ can be comparable across neighboring layers, supporting structural analysis of routing behavior.

To quantify routing smoothness, we define **pathway consistency** $C_i$ as the normalized average cosine similarity between consecutive layer embeddings for input $x_i$:

$$C_i = 1 - \frac{1}{L-1} \sum_{\ell=1}^{L-1} \frac{\cos(\mathbf{e}_{i,\ell}, \mathbf{e}_{i,\ell+1})}{\max_\ell \cos(\mathbf{e}_{i,\ell}, \mathbf{e}_{i,\ell+1}) + \epsilon},$$

where $\epsilon = 10^{-8}$ ensures numerical stability. Higher $C_i$ indicates more erratic transitions, while lower values reflect smoother, more consistent routing across layers.

Figure 6 tracks the evolution of pathway consistency and pretraining objective across four domains. A key finding is that even after the objective stabilizes, pathway consistency continues to improve—revealing ongoing refinement in the internal routing of the model. This suggests that smoother, more coherent cross-layer transitions emerge well after memorization, positioning pathway consistency as a valuable lens for understanding generalization beyond what the training objective alone captures.

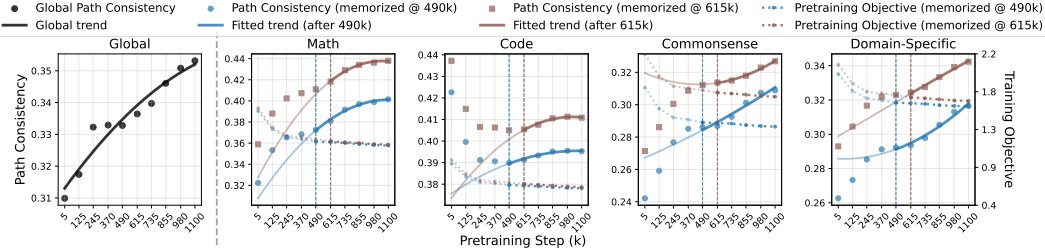

Figure 6: **Pathway consistency vs. training objective** on four domains during pretraining. The global panel (leftmost) shows that the path consistency consistently improves over all data, while each domain panel shows the same trend on two data groups (memorized at $490k$ and $615k$) from each domain after the training objective converges. This reflects a progressively smoother transition of experts between consecutive layers after memorization.

## 4.3 PATHWAY COMPLEXITY METRICS AS MONITORS OF GENERALIZATION

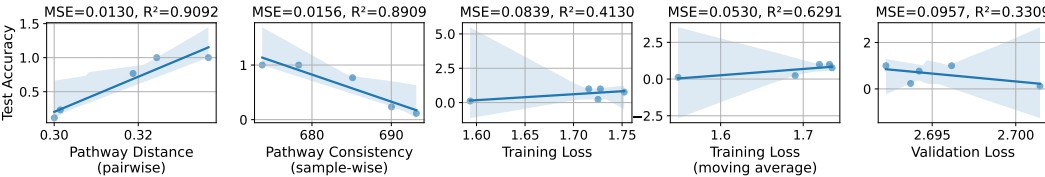

Figure 7: Correlation of the two pathway complexity metrics with benchmark performance. Compared to the training objective and its moving average, both the pathway edit distance and consistency are highly correlated with the test accuracy, though they are also computed on training data.

Table 1: **Correlation between pathway complexity metrics and generalization** (test accuracy) across four domains, with the comparison to training/validation loss. $p$-value: $p < 0.1$ , $p < 0.05$ , $p < 0.01$ , $p < 0.001$ , $p > 0.1$ , $p > 0.5$ , $p > 0.8$ . Preferred correlation: positive , negative .

| Metric | Math | | Code | | Commonsense | | Domain-Specific | |
|---|---|---|---|---|---|---|---|---|
| | Pearson | Spearman | Pearson | Spearman | Pearson | Spearman | Pearson | Spearman |
| Pathway Distance (pairwise) | -0.9471 | -0.9487 | -0.9290 | -0.9276 | -0.9821 | -0.9747 | -0.9254 | -0.9487 |
| Pathway Consistency (sample-wise) | 0.9246 | 0.9487 | 0.9786 | 0.9856 | 0.9804 | 0.9747 | 0.9917 | 0.9487 |
| Training Loss | -0.0435 | -0.2108 | -0.0680 | -0.2052 | 0.6427 | 0.4104 | 0.7944 | 0.9487 |
| Training Loss (moving average) | 0.4714 | 0.3162 | 0.2481 | -0.2052 | 0.7931 | 0.6669 | 0.8705 | 0.9487 |
| Validation Loss | 0.4801 | -0.1054 | 0.5466 | 0.8721 | -0.5752 | -0.4617 | -0.3339 | -0.3162 |

Given the strong link between routing dynamics and generalization, we ask: *To what extent do our pathway metrics predict test performance across domains?* To answer this, we compute Pearson and Spearman correlations between downstream performance and two metrics across pretraining checkpoints. As described in Section 3.1, we apply a lightweight LoRA-based instruction tuning (rank = 32; see Appendix B) solely to equip pretrained checkpoints with basic instruction-following ability for benchmark evaluation. The robustness of this setup is further supported by the LoRA rank ablation in Appendix E.2. Our correlation analysis focuses on checkpoints after domain-level accuracy begins its sustained rise (shown in Figure 1), thereby isolating the mechanisms that drive generalization beyond the random-accuracy phase.

Table 1 summarizes the correlation results across four tasks. Pathway consistency (sample-wise) shows exceptionally strong positive correlations with test accuracy—often exceeding $0.97$ and highly significant—indicating that coherent, stable routing across layers is a key marker of generalization. In contrast, pairwise pathway similarity consistently yields strong negative correlations (around -0.93), suggesting that as generalization improves, inputs tend to follow increasingly similar expert routes. By comparison, as shown in Figure 7, training and validation loss metrics exhibit weaker and less consistent correlations. Notably, while moving-average training loss shows relatively high correlation in certain domains (e.g., Code, Domain-Specific), its direction is inconsistent with expectation: since loss is minimized during training, we would expect a *negative* correlation with test accuracy—not the *positive* one observed here. This mismatch further underscores that conventional pretraining metrics fail to reflect the internal transition from memorization to generalization.

These findings position pathway metrics as robust, domain-general indicators of generalization. Importantly, they rely only on training dynamics—without requiring held-out validation data—making them especially valuable for real-time monitoring in large-scale or resource-constrained settings.

### 4.4 THEORETICAL CONNECTIONS & EXPLANATIONS

We connect expert routing patterns with model generalization via the *complexity* of routing, formalized through the *effective dimension* of the routing kernel. This effective dimension will serve as the key determinant of MoE generalization.

Concretely, $\boldsymbol{H}^*$ is the Gram matrix of the routing kernel with entries $\boldsymbol{H}_{ij}^* = \Theta_{\text{route}}(x_i, x_j)$, capturing similarity between inputs induced by both routing overlap and expert functions. The effective dimension is defined as $\text{Tr}(\boldsymbol{H}^*(\boldsymbol{H}^* + \lambda I)^{-1})$, measuring the complexity of the function space induced by routing.

**Theorem 4.1** (Generalization Bound). *Under NTK regime with $K$ experts, with probability $1 - \delta$ over $n$ samples, where $C_1, C_2$ are NTK-dependent constants and $\lambda$ is the regularization parameter:*

$$\mathcal{E} \leq \underbrace{\frac{C_1\lambda^2}{n}\boldsymbol{y}^{\top}(\boldsymbol{H}^* + \lambda I)^{-2}\boldsymbol{y}}_{Bias} + \underbrace{C_2\left(\frac{\sigma^2 Tr(\boldsymbol{H}^*(\boldsymbol{H}^* + \lambda I)^{-1})}{n} + \frac{\sigma^2 \log(1/\delta)}{n}\right)}_{Variance + Noise}$$

This provides a lens on grokking: as routing evolves from random to structured, the effective dimension collapses, triggering improvements in generalization. Intuitively, this collapse corresponds to a simplification of routing pathways: the kernel spectrum becomes more concentrated, reducing the hypothesis space and mitigating overfitting. The full theoretical setup, assumptions, and proof are detailed in Appendix C, while Appendix C.4 empirically confirms that effective dimension is

tightly aligned with routing edit distance, establishing pathway complexity as a key driver of MoE generalization.

## 5 DISCUSSION

**Why studying OLMoE matters.**  OLMoE is the only publicly available large-scale MoE LLM with full pretraining checkpoints, providing a rare opportunity to study pretraining dynamics directly. Its architecture, including top-$k$ routing, load-balancing loss, AdamW, RMSNorm, and RoPE, follows standard MoE practices and thus reflects typical large-scale designs. While additional pretraining checkpoints are not yet available, recent studies on other MoE models such as DeepSeekMoE (Dai et al., 2024) and Qwen3-MoE (Yang et al., 2025) show that promoting similar routing pathways across related samples improves generalization through post-training (Li et al., 2025c) or test-time adaptation (Li et al., 2025a;b). These findings support our conclusion that pathway similarity is a general signal of model generalization and that the mechanisms observed in OLMoE extend to other MoE architectures.

**Ruling out global routing effects.**  To ensure that the observed decrease in pathway distance is not driven by global factors such as load balancing, we conduct two analyses. First, a cross-domain comparison (Appendix E.4) shows that pathway distances between different domains (e.g., math $\times$ code) fluctuate irregularly without a consistent downward trend, in contrast to the observed group- and domain-specific decreases in Figure 4, confirming that pathway reorganization is a localized phenomenon driven by memorization rather than a global routing effect. Second, the correlation between routing entropy and pathway edit distance is weak ($r = 0.24$, ∼6% variance explained; Appendix E.5), indicating that balanced or saturated routing alone does not account for our findings. Together, these results demonstrate that pathway reorganization arises from structured, domain-specific computation rather than global changes in routing behavior.

**Consistent Findings on another MoE (nanoMoE) pretraining justify the generality of our study.**  To further assess the generality of our findings, we train a separate MoE model from scratch (nanoMoE; Appendix E.6) to obtain fully observable pretraining dynamics. Despite differences in scale, data, and training setup, the model exhibits the same post-memorization routing behavior, where pathway distance decreases and pathway consistency increases, and both metrics show strong correlations with downstream benchmark performance. This independent experiment confirms that the pathway–generalization relationship is a robust property of MoE training dynamics rather than an artifact of OLMoE.

## 6 CONCLUSION

This paper provides the first empirical evidence that, in large-scale LLM pretraining, generalization can emerge well after memorization has been achieved. Through an analysis of routing dynamics in a 7B-parameter MoE model, we propose two novel pathway-based metrics that accurately track generalization without requiring finetuning or external validation sets. Our findings reveal a transition from memorization to generalization, marked by increased pathway similarity and consistency, and supported by theoretical connections between routing structure and generalization bounds. These insights offer a foundation for more transparent and reliable monitoring of generalization in large-scale models. Future work will extend these pathway-based metrics beyond MoE, by constructing analogous "virtual pathways" in dense models, moving toward a unified framework for tracking generalization in foundation models.

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

## A  DATASET DETAILS

We evaluate OLMoE on four domains used during pretraining: **math**, **code**, **commonsense**, and **domain-specific** (scientific text). For each domain, we construct three sets:

- **Pretraining sample set:** 10,000 examples drawn from OLMoE's pretraining data.
- **Test sample set:** 10,000 examples from a benchmark dataset in the same domain, following OLMoE's categorization where available.
- **Validation set:** a disjoint dataset used solely to compute validation loss, enabling correlation analysis between in-domain learning and test generalization.

For the domain-specific case, we filter the scientific pretraining data using the keyword *"Organic Chemistry"*, aligning the training distribution with the "college_chemistry" benchmark in MMLU.

Table 2 summarizes all dataset sources. All pretraining sets are directly drawn from OLMoE's pretraining corpora; test benchmarks are standard evaluation datasets; validation sets are only used for correlation analysis.

Table 2: Datasets used in our domain-level evaluation. Pretraining samples are from OLMoE corpora, test benchmarks are used for evaluation, and validation sets are disjoint corpora used only for loss–accuracy correlation. All datasets accessed via HuggingFace.

| Domain | Pretraining Source | Test Dataset | Validation Dataset |
|---|---|---|---|
| Math | `open-web-math/` `open-web-math` | `hkust-nlp/` `dart-math-pool-math` | `qwedsacf/` `competition_math` |
| Code | `bigcode/` `starcoderdata` | `evalplus/` `humanevalplus` | `newfacade/` `LeetCodeDataset` |
| Commonsense | `emozilla/` `dolma-v1_7-books` | `tau/commonsense_qa` | `kowndinya23/` `flan2021-commonsense` |
| Domain-Specific | `blester125/` `mediawiki-dolma` (filtered "Organic Chemistry") | `cais/mmlu` ("college_chemistry" subset) | `chemNLP/` `chemistry-bookshelves` `-merged` |

## B  EXPERIMENT DETAILS

**Compute Resources.** All experiments are conducted on NVIDIA A100 GPUs with 80GB memory, using public OLMoE checkpoints.

**Instruction Tuning.** The raw OLMoE checkpoints exhibit weak instruction-following ability, making direct evaluation of generalization behavior unreliable—particularly in text-based domains such as commonsense reasoning or code completion. To address this, we apply light instruction tuning using LoRA (Hu et al., 2022) to make the pretrained model *follow prompts in a human-readable way*. This ensures we can evaluate generalization while preserving the original pretraining behavior.

To avoid overriding pretraining representations, we adopt minimal finetuning using a small number of epochs and low-rank parameter updates. Specifically:

- **Math:** Finetune with LoRA for 3 epochs on the same math SFT dataset `meta-math/` `MetaMathQA` used by OLMoE.
- **Code:** Finetune for 3 epochs on `HuggingFaceH4/CodeAlpaca_20K`.
- **Commonsense & Domain-Specific:** Finetune for 3 epochs on `yahma/alpaca-cleaned`.

Key configurations of LoRA include:

- **Target modules:** `q_proj`, `k_proj`, `v_proj`
- **LoRA rank:** 32
- **LoRA dropout:** 0.1

- **Learning rate:** 5e-5
- **Batch size:** 4
- **Max sequence length:** 2048

**Applying Min-K%++ for Training-Test Separation.** To maintain a rigorous separation between the pretraining and test datasets, we employ the Min-K%++ membership inference attack (Zhang et al., 2024), a state-of-the-art, reference-free method designed to detect whether specific data samples are part of a model's training corpus.

Specifically, we apply the Min-K%++ method to the test dataset to identify samples that the model is most likely to have encountered during training. Determining an optimal threshold for classification is challenging due to variability in model behaviors and data distributions. To address this, we adopt a conservative approach by removing the top 10% of test samples with the highest Min-K%++ scores, thereby minimizing the risk of training data contamination in the test set.

This strategy ensures that our evaluation metrics accurately reflect the model's generalization capabilities on truly unseen data, enhancing the validity of our experimental results.

### B.1 CONSTRUCTION OF TRAINING–TEST PAIRED GROUPS.

Here we provide additional details on how we construct the training–test paired groups used in Figure 2 to analyze the temporal relation between memorization and generalization. The overall procedure is introduced in Section 3.2; here, we summarize the specific grouping and matching setup for clarity and reproducibility.

**Training groups.** For each pretraining sample, we track its token-level loss across checkpoints and mark the earliest step $t_i^*$ when the loss remains below a threshold $\varepsilon$ and deviates by less than $\delta$ from its final value for all later checkpoints. Samples sharing the same $t_i^*$ form a *training group* denoted as *Pretrain@$t_i^*$*. For $(\varepsilon, \delta)$, $\delta$ is fixed at 0.05 across domains, while $\varepsilon$ is domain-specific and determined based on the average loss scale of each domain. We test $\varepsilon$ in increments of 0.1 to ensure that the cumulative proportion of memorized samples before the final checkpoint remains between 20–25% across domains, balancing sufficient data coverage with stable per-domain scaling. Ablation studies on the robustness of $(\varepsilon, \delta)$ is provided in Appendix E.3.

**Test groups.** Each test sample's accuracy trajectory is recorded across checkpoints. A test sample is considered to have generalized at the earliest step $t_j^{\#}$ where its accuracy remains consistently correct thereafter, with mean accuracy above 0.8 and at most one error in the following checkpoints. Samples sharing the same $t_j^{\#}$ form a *test group* denoted as *Test@$t_j^{\#}$*.

**Embedding-based pairing.** To align semantically related training and test groups, we compute sentence embeddings for all samples using the SentenceTransformer model "all-MiniLM-L6-v2" and measure pairwise cosine similarity between groups within each domain. The average similarity between every pair of groups forms a similarity matrix, and a one-to-one mapping between training and test groups is obtained using the Hungarian algorithm implemented in SciPy, which maximizes overall semantic similarity. The notation *Pretrain@$p$ → Test@$q$* thus indicates that the training group memorized at step $p$ is semantically aligned with a test group whose performance stabilizes at step $q$.

## C GENERALIZATION BOUND FOR MoE WITH ROUTING KERNEL

### C.1 MODEL SETUP AND ASSUMPTIONS

Consider a MoE model where the router is fixed after pretraining, and the expert networks are trainable. Let $K$ be the total number of experts in the model.

- $f_k(\phi_k, x) \in \mathbb{R}$ be the $k$-th expert's output, with trainable parameters $\phi_k$.
- $g_k(x) \in [0, 1]$ be the $k$-th routing weight, which is fixed and pre-determined for any input $x$. We assume $\sum_{k=1}^{K} g_k(x) = 1$ (e.g., from a fixed softmax layer or pre-computed weights).

- **Full model:** $F(\theta, x) = \sum_{k=1}^{K} g_k(x) f_k(\phi_k, x)$, where $\theta = (\phi_1, \ldots, \phi_K)$ is the collection of all trainable expert parameters.

**Key Assumptions:**

1. **NTK Regime for Experts:** Expert parameters $\theta = (\phi_1, \ldots, \phi_K)$ are initialized as $\phi_k(0) \sim \mathcal{N}(0, I)$ for each $k$, and remain $\epsilon$-close to initialization ($\|\theta(t) - \theta(0)\| = O(1/\sqrt{\text{width}})$), enabling linearization via Neural Tangent Kernel (NTK). We assume $\phi_k(0)$ are independent across different experts $k$.

2. **Fixed Routing:** The routing weights $g_k(x)$ are fixed for all inputs $x$ and do not contain any trainable parameters. They satisfy $\max_k \|g_k\|_\infty \leq 1$.

3. **Label Noise:** Training labels $y_i = f^*(x_i) + \epsilon_i$ with $\epsilon_i \sim \mathcal{N}(0, \sigma^2)$, independent of $x_i$.

## C.2 ROUTING KERNEL DEFINITION

Define the NTK for this MoE configuration. Consistent with the general routing kernel definition in the main text, $\Theta_{\text{route}}(x, x')$ here is specifically instantiated as:

$$\Theta_{\text{route}}(x, x') = \mathbb{E}_{\theta(0)} \left[ \left\langle \frac{\partial F(\theta(0), x)}{\partial \theta}, \frac{\partial F(\theta(0), x')}{\partial \theta} \right\rangle \right]$$

$$= \sum_{j=1}^{K} g_j(x) g_j(x') \mathbb{E}_{\phi_j(0)} \left[ \left\langle \frac{\partial f_j(\phi_j(0), x)}{\partial \phi_j}, \frac{\partial f_j(\phi_j(0), x')}{\partial \phi_j} \right\rangle \right]$$

Let $K_{f_j}(x, x') = \mathbb{E}_{\phi_j(0)} \left[ \left\langle \frac{\partial f_j(\phi_j(0), x)}{\partial \phi_j}, \frac{\partial f_j(\phi_j(0), x')}{\partial \phi_j} \right\rangle \right]$ be the NTK for the $j$-th expert network. Then, the Routing Kernel in this fixed-routing scenario is:

$$\Theta_{\text{route}}(x, x') = \sum_{j=1}^{K} g_j(x) g_j(x') K_{f_j}(x, x')$$

The structure of $\Theta_{\text{route}}(x, x')$ in this fixed-routing setup provides crucial insights into how routing influences generalization. It is composed of two primary factors: the gating weight product $g_j(x) g_j(x')$, and the individual expert NTKs $K_{f_j}(x, x')$. The term $g_j(x) g_j(x')$ signifies the importance of the $j$-th expert for both inputs $x$ and $x'$, highlighting that two inputs are considered "similar" by the kernel if they are significantly routed to the *same* experts. If inputs $x$ and $x'$ are routed to distinct sets of experts, their contribution to the sum for a given $j$ will be diminished. Simultaneously, $K_{f_j}(x, x')$ captures the functional similarity learned by the $j$-th expert for inputs $x$ and $x'$. Thus, the overall routing kernel measures a compounded similarity: inputs are similar not only if they are guided through similar pathways (weighted by $g_j(x) g_j(x')$), but also if the specific experts they pass through exhibit similar functional behavior for those inputs ($K_{f_j}(x, x')$). This decomposition underscores how fixed routing patterns (reflected in $g_j(x)$) set the stage for expert specialization and ultimately shape the generalization properties of the MoE model.

## C.3 MAIN THEOREM

**Theorem C.1** (Generalization Bound for MoE Experts). *Under Assumptions 1-3, with probability $1 - \delta$ over training samples $(x_i, y_i)_{i=1}^{n}$, the excess risk satisfies:*

$$\mathbb{E}_{x,y} \left[ (F(\theta^*, x) - f^*(x))^2 \right] \leq \underbrace{\frac{C_1 \lambda^2}{n} \boldsymbol{y}^\top (\boldsymbol{H}^* + \lambda I)^{-2} \boldsymbol{y}}_{Bias}$$

$$+ \underbrace{C_2 \left( \frac{\sigma^2 Tr(\boldsymbol{H}^*(\boldsymbol{H}^* + \lambda I)^{-1})}{n} + \frac{\sigma^2 \log(1/\delta)}{n} \right)}_{Variance + Noise}$$

*where $\boldsymbol{H}_{i,j}^* = \Theta_{route}(x_i, x_j)$, $\lambda > 0$ is regularization, and $C_1, C_2$ are constants depending on the properties of $g_k(x)$ and $f_k(x)$. The bound decays as $O\left(\frac{1}{n}\right)$ when $\lambda = \Theta(1)$.*

*Proof.* Under NTK regime, the model can be linearized around initialization:

$$F(\theta, x) \approx F(\theta(0), x) + \langle \nabla_\theta F(\theta(0), x), \theta - \theta(0) \rangle$$
$$= \underbrace{F(\theta(0), x)}_{\text{Initial Output}} + \Phi(x)^\top \Delta\theta$$

where $\Phi(x) = \nabla_\theta F(\theta(0), x)$ and $\Delta\theta = \theta - \theta(0)$. The quadratic error term $\|\Delta\theta\|^2$ is $O(1/\text{width})$ and thus negligible.

Define centered labels $\tilde{y}_i = y_i - F(\theta(0), x_i)$. The problem reduces to ridge regression:

$$\min_{\Delta\theta} \frac{1}{2} \sum_{i=1}^n \left( \Phi(x_i)^\top \Delta\theta - \tilde{y}_i \right)^2 + \frac{\lambda}{2} \|\Delta\theta\|^2$$

with closed-form solution:

$$\Delta\theta^* = (\Phi\Phi^\top + \lambda I)^{-1} \Phi\tilde{\boldsymbol{y}} = \Phi^\top (\boldsymbol{H}^* + \lambda I)^{-1} \tilde{\boldsymbol{y}}$$

where $\boldsymbol{H}^* = \Phi\Phi^\top$ is the kernel matrix with $\boldsymbol{H}^*_{i,j} = \Theta_{\text{route}}(x_i, x_j)$.

The excess risk decomposes as:

$$\mathbb{E}[(F(\theta^*, x) - f^*(x))^2] = \underbrace{(\mathbb{E}[\hat{f}(x)] - f^*(x))^2}_{\text{Bias}^2} + \underbrace{\mathbb{E}[(\hat{f}(x) - \mathbb{E}[\hat{f}(x)])^2]}_{\text{Variance}} + \sigma^2$$

- **Bias Term**:

$$\text{Bias}^2 \leq \frac{\lambda^2}{n^2} \boldsymbol{y}^\top (\boldsymbol{H}^* + \lambda I)^{-2} \boldsymbol{y} \leq \frac{C_1 \lambda^2}{n} \boldsymbol{y}^\top (\boldsymbol{H}^* + \lambda I)^{-2} \boldsymbol{y}$$

where $C_1$ absorbs constants related to the magnitudes of $g_k(x)$ and expert gradients.

- **Variance Term**: The parameter covariance is $\text{Cov}(\Delta\theta^*) = \sigma^2 (\Phi\Phi^\top + \lambda I)^{-1} \Phi\Phi^\top (\Phi\Phi^\top + \lambda I)^{-1}$. For prediction variance:

$$\text{Variance} = \mathbb{E}_x[\Phi(x)^\top \text{Cov}(\Delta\theta^*) \Phi(x)] = \frac{\sigma^2}{n} \text{Tr}(\boldsymbol{H}^* (\boldsymbol{H}^* + \lambda I)^{-1})$$

- **Confidence Term**: By standard concentration inequalities (e.g., for sub-Gaussian noise),

$$\mathbb{P}\left( \left| \frac{1}{n} \sum_{i=1}^n \epsilon_i^2 - \sigma^2 \right| \geq t \right) \leq e^{-nt^2/(2\sigma^4)} \implies \frac{\sigma^2 \log(1/\delta)}{n}$$

Collecting all components and adjusting constants $C_1, C_2$ completes the proof. $\square$

## C.4 CONNECTING EDIT DISTANCE TO EFFECTIVE DIMENSION

The bound shows that simple (low-dimensional) routing kernels generalize better. We now empirically test how routing pattern complexity (via edit distance) correlates with this effective dimension $d_{\text{eff}} = \text{Tr}(\boldsymbol{H}^* (\boldsymbol{H}^* + \lambda I)^{-1})$.

Our setup involves a lightweight MoE architecture with 16 experts and a 100-dimensional input space. Input data are synthetic Gaussian clusters, providing a controlled environment to analyze pre-defined routing strategies. For each experimental trial, 200 samples are split into training and validation sets. We train a one-layer MoE model with ReLU activations where only expert networks are trainable; the router's weights are initialized once and then frozen. To explore diverse fixed routing patterns, we perform multiple independent runs, *each starting with a different random initialization of the router*.

Following training, we extract the average edit distance of the pre-determined expert routing paths and the effective dimension. As illustrated in Figure 8, we observe a strong positive correlation between these two quantities. This empirically demonstrates that the structural properties of fixed routing, measured by edit distance, meaningfully impact the complexity of the function space learned by the experts, as predicted by $d_{\text{eff}}$. This alignment supports edit distance as a valuable diagnostic for MoE generalization.

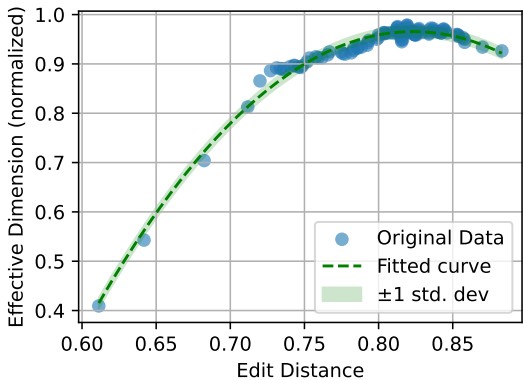

Figure 8: **Correlation of Edit Distance with Effective Dimension.** Each point represents an independent experimental run where the router is initialized with a different random seed and then fixed for expert training. This varying initialization leads to diverse fixed routing patterns. We observe a strong positive correlation. This suggests that the structural diversity of pre-defined routing significantly impacts the complexity of the function space learned by experts.

## D    ROUTING DYNAMICS ANALYSIS

To investigate whether routing evolution arises primarily from router-layer updates or other weight updates, we analyze convergence speed and stability across parameter types.

**Results.**    Table 3 reports convergence speed (magnitude drop-off over time) and stability (consistency of updates). Router weights adapt fastest while expert weights stabilize most gradually, indicating a division of labor: routing decisions are steered by fast-updating routers atop slow-evolving expert features. Attention layers lie in between.

Table 3: Convergence speed and stability across parameter types. Higher values indicate faster convergence or greater stability.

| Parameter Type | Convergence Speed ↑ | Stability ↑ |
| --- | --- | --- |
| Router | **0.151** (fastest) | 5.63 |
| Expert | 0.063 (slowest) | **7.32** |
| Attention | 0.083 | 6.01 |

**Layerwise Routing Dynamics.**    To localize routing evolution, we compute the edit distance of routing assignments across layers. Layers 0–3 exhibit the most post-convergence change, suggesting that early layers remain active in refining pathways.

**Metric Computation.**    We periodically snapshot parameter L2 norms during training. For each parameter type, we compute the mean absolute norm change between checkpoints, normalized by average scale. *Convergence speed* is defined as the drop in change magnitude from early to late training, while *stability* is the inverse standard deviation of these normalized changes. This simple but robust analysis consistently reveals router–expert asymmetries.

## E    ANALYSIS

### E.1    EFFECT OF ROUTING-WEIGHT THRESHOLD ON PATHWAY DYNAMICS

To examine the robustness of our findings with respect to the routing-weight threshold used for pathway construction, we repeat the analysis in Fig. 4 under different cumulative routing-weight

thresholds ranging from 0.5 to 0.8 (interval 0.1). This threshold determines how many experts are considered *active* in each layer when forming the pathway sequence: a higher threshold includes more experts, while a lower threshold selects only the most dominant ones.

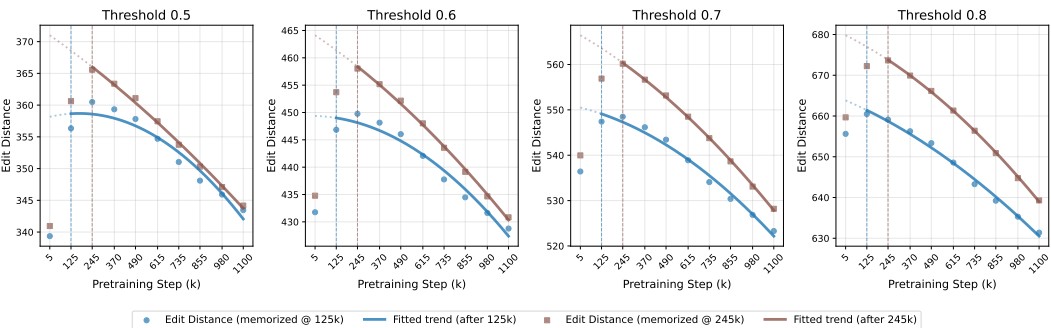

Figure 9: **Effect of routing-weight threshold on pathway edit distance on domain-specific QA tasks.** Each panel shows the average pathway edit distance for this domain under a different cumulative routing-weight threshold (0.5–0.8, interval 0.1). Higher thresholds include more active experts, leading to longer absolute distances, but all settings exhibit the same qualitative pattern: pathway edit distance decreases consistently after memorization, confirming that the observed dynamics are robust to the threshold choice.

Figure 9 presents the results for the domain-specific QA tasks. Although the absolute values of edit distance increase with larger thresholds—reflecting the longer sequences formed when more experts are included—the qualitative trends remain consistent across all settings. For every threshold, the pathway edit distance decreases after memorization (marked by vertical dashed lines), and the decrease becomes more pronounced at later checkpoints.

These results demonstrate that our conclusions are **robust to the choice of routing-weight threshold**. The observed reduction in pathway complexity after memorization, corresponding to the **emergence of shared pathways**, persists across all threshold settings.

### E.2    EFFECT OF LoRA RANK ON PATHWAY–GENERALIZATION CORRELATION

To examine whether the correlation between pathway metrics and generalization depends on the configuration of instruction tuning, we conduct an ablation on the LoRA rank used during lightweight finetuning. As described in Appendix B, the default setup adopts a rank of 32 for all domains. Here, we additionally train a LoRA adapter with rank 64 on the *commonsense* domain, following the same data, training steps, and hyperparameters as in the main experiments.

Table 4 reports the correlation between pathway complexity metrics and downstream test accuracy under the two LoRA ranks. The results show that increasing the rank from 32 to 64 yields nearly identical correlations for both *Pathway Distance* and *Pathway Consistency*, while training and validation losses remain weakly correlated. This consistency indicates that the observed relationships between pathway metrics and generalization are **robust to LoRA configuration** and primarily reflect intrinsic pretraining-phase dynamics rather than artifacts of instruction tuning.

### E.3    EFFECT OF MEMORIZATION THRESHOLDS $\varepsilon$ AND $\delta$

We conduct an ablation study on the two hyperparameters that define the memorization criterion in Section 3.1: the loss threshold $\varepsilon$ and the stability threshold $\delta$. Specifically, for a sample $x_i$ with loss $\ell_i(\theta_t)$ at pretraining step $t$, it is considered *memorized* from step $t_i^*$ if $\ell_i(\theta_t) \leq \varepsilon$ and $|\ell_i(\theta_t) - \ell_i(\theta_T)| \leq \delta$ for all $t \geq t_i^*$. We evaluate how sensitive the pathway trends are to these two parameters using the domain-specific QA task.

**Varying $\varepsilon$.**    Figure 10 shows results when fixing $\delta = 0.05$ and varying $\varepsilon$ from 1.5 to 2.0. Across all settings, the pathway edit distance after memorization consistently decreases with training, and the

Table 4: **Effect of LoRA Rank on the Correlation Between Pathway Metrics and Generalization** (Commonsense reasoning domain). The results show that increasing the LoRA rank from 32 to 64 yields nearly identical correlations, confirming that the observed trends are robust to LoRA configuration. $p$-value: $p < 0.1$ , $p < 0.05$ , $p < 0.01$ , $p > 0.1$ , $p > 0.5$ , $p > 0.8$ . Preferred correlation: positive , negative .

| Metric | LoRA Rank = 32 | | LoRA Rank = 64 | |
|---|---|---|---|---|
| | Pearson | Spearman | Pearson | Spearman |
| Pathway Distance (pairwise) | -0.9821 | -0.9747 | -0.9927 | -0.9487 |
| Pathway Consistency (sample-wise) | 0.9804 | 0.9747 | 0.9559 | 0.9487 |
| Training Loss | 0.6427 | 0.4104 | 0.6197 | 0.4617 |
| Training Loss (moving average) | 0.7931 | 0.6669 | 0.6871 | 0.2052 |
| Validation Loss | -0.5752 | -0.4617 | -0.6631 | -0.4104 |

overall trend remains stable—indicating that $\varepsilon$ only shifts the point where samples are considered memorized but does not affect the post-memorization dynamics.

**Varying $\delta$.** Figure 11 reports the results when fixing $\varepsilon = 2.0$ and varying $\delta$ between 0.03 and 0.05. Similarly, the decreasing trend of edit distance after memorization persists, and the fitted trajectories show negligible variation across $\delta$ values.

This confirms that the proposed pathway-based observations are **robust to the choice of $\varepsilon$ and $\delta$**.

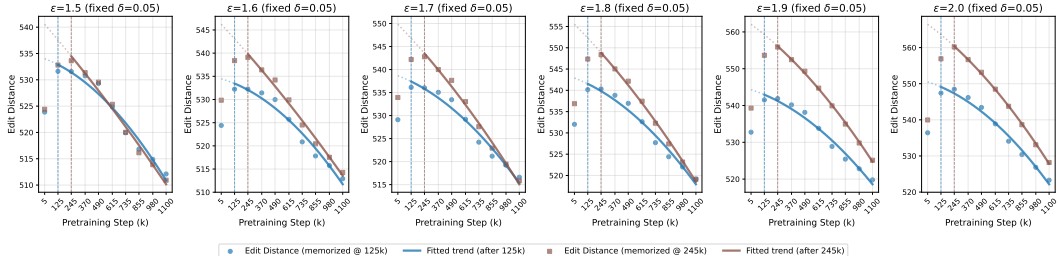

Figure 10: **Effect of the loss threshold $\varepsilon$ on pathway dynamics.** The stability threshold is fixed at $\delta = 0.05$. All curves show a consistent post-memorization decrease in edit distance, indicating that the results are robust to $\varepsilon$.

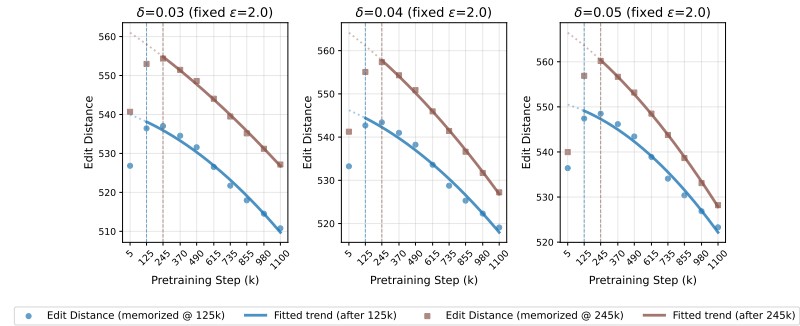

Figure 11: **Effect of the stability threshold $\delta$ on pathway dynamics.** The loss threshold is fixed at $\varepsilon = 2.0$. The same decreasing trend holds across different $\delta$ values, confirming robustness to this parameter.

### E.4 CROSS-DOMAIN ANALYSIS OF PATHWAY DYNAMICS

To verify that the observed changes in pathway edit distance are due to the emergence of shared pathways within specific domains rather than uniform effects across all data, such as load balancing

or regularization, we conduct an additional analysis focusing on the *cross-domain* samples. Using the same cumulative routing-weight threshold (0.7) as in the main experiments, we compute average pathway edit distances for samples drawn from two different domains across checkpoints—specifically, math $\times$ code—under identical configurations.

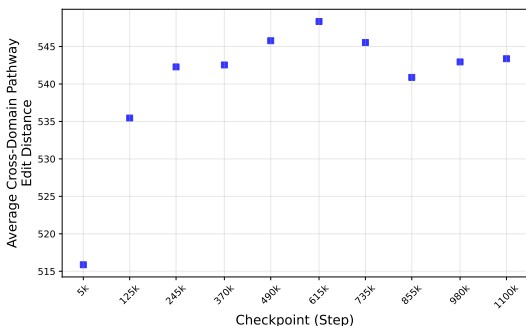

Figure 12: **Cross-domain pathway edit distance (math $\times$ code).** The figure shows the global trend of average pathway edit distance during pretraining when computed across two different domains. The cross-domain distances fluctuate irregularly, exhibiting both increases and decreases but no systematic downward trend. This contrasts with the localized decreases observed within domains (Fig. 4), indicating that pathway reorganization occurs locally among semantically related data rather than globally across domains.

As shown in Fig. 12, the cross-domain edit distance fluctuates throughout pretraining, showing irregular increases and decreases without any consistent downward trend. In contrast, Fig. 4 demonstrates that within-domain pathway distances decrease only for groups of samples that become memorized at different points in training, with each domain exhibiting its own localized transition rather than a global drop. These observations confirm that the structural changes we observe are **domain- and group-specific**, reflecting the **emergence of shared pathways among semantically related samples** rather than a global effect of routing behavior.

### E.5 RELATIONSHIP BETWEEN ROUTING ENTROPY AND EDIT DISTANCE

We investigate whether changes in routing diversity could trivially explain the observed trends in pathway edit distance. Specifically, we test whether routing entropy—high when expert usage is balanced and low when a few experts dominate—can drive edit distance dynamics.

For each sample, routing entropy is computed *per layer* from the full routing-weight vector (no thresholding or selection) and averaged across layers. For each sample pair, we take the mean of the two samples' entropies and compute their pathway edit distance using a cumulative routing-weight threshold of 0.8, which provides sufficient expert coverage for analysis.

We sample 141,530 pairs across checkpoints spanning both the *math* and *code* domains, including within-domain and cross-domain pairs. Figure 13 shows a weak correlation between average routing entropy and edit distance ($r = 0.24$, explaining about $6\%$ of the variance) with no clear monotonic or directional trend. This indicates that neither more balanced routing (higher entropy) nor more concentrated routing (lower entropy) systematically reduces pathway distance. Therefore, the decrease in within-domain pathway distance observed after memorization is unlikely to result from overall changes in routing balance; instead, it reflects domain- and group-specific reorganization of computation.

### E.6 CONSISTENT FINDINGS ON ANOTHER MOE (NANOMOE)

To further validate the generality of our findings beyond OLMoE, we conduct additional experiments on a 55M-parameter MoE model, **nanoMoE**[1]. NanoMoE provides a small, fully trainable MoE architecture for which we can obtain complete pretraining checkpoints, making it suitable for analysis of routing dynamics.

---

[1] https://github.com/wolfecameron/nanoMoE

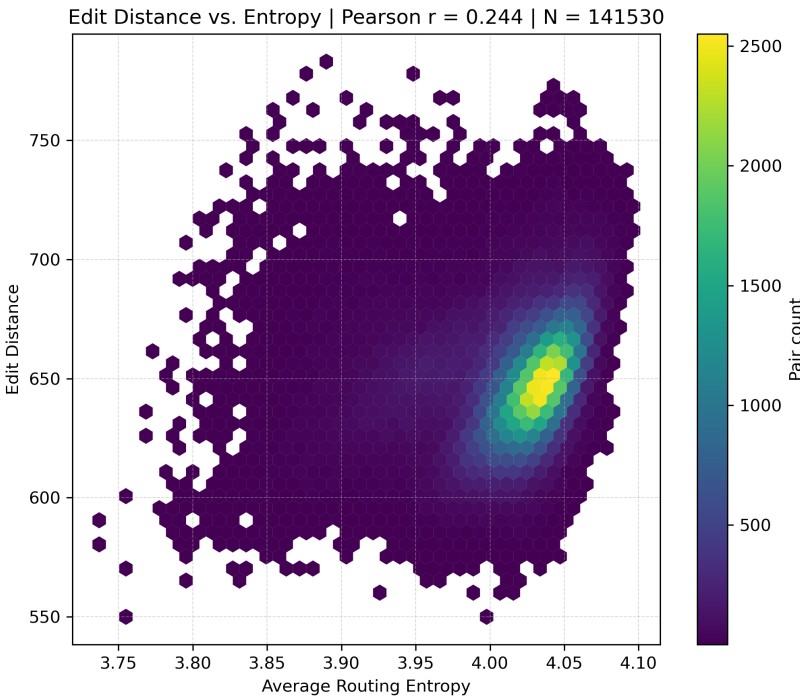

Figure 13: **Edit distance vs. routing entropy.** Each hexagon aggregates sample pairs ($N = 141,530$) drawn across checkpoints and across the *math* and *code* domains. Routing entropy is computed from the *full* routing-weight vectors (no thresholding); pathway edit distance uses pathways built with a cumulative cutoff of $0.8$. The weak correlation ($r = 0.24$) indicates that routing balance does not systematically determine pathway similarity.

**Training setup.** We train nanoMoE on ∼25B tokens of the OpenWebText dataset (Gao et al., 2020) for 50k iterations on two NVIDIA H100 GPUs and uniformly sample ten checkpoints for analysis. OpenWebText is a mixed-domain web corpus with a heavy emphasis on commonsense and general knowledge content. To approximate a domain setting, we construct a commonsense subset by filtering 10,000 randomly sampled training samples using Qwen3-8B (Yang et al., 2025) to select those semantically related to commonsense reasoning, yielding 7,857 samples in total. Following the same methodology as our main experiments, we identify memorized samples based on token-level loss stability and compute pathway dynamics for this subset across checkpoints.

**Pathway-metric dynamics.** Consistent with our large-scale results, we observe that after memorization, **pathway edit distance decreases while pathway consistency increases** as shown in Figure 14 and 15. These trends mirror the post-memorization reorganization we identify in OLMoE, demonstrating that even the smaller MoE model reorganizes routing toward more structured, shared computation after the loss has already converged.

**Correlation with downstream performance.** We further evaluate all ten checkpoints in a zero-shot setting on standard commonsense reasoning benchmarks (ARC-Easy/Challenge (Clark et al., 2018), HellaSwag (Zellers et al., 2019), and OpenBookQA (Mihaylov et al., 2018)). Zero-shot evaluation is justified because models pretrained on OpenWebText, including GPT-2 (Radford et al., 2019), have been shown to achieve strong zero-shot performance on commonsense benchmarks. As in the large-scale setting, both pathway metrics exhibit strong correlations with benchmark accuracy ($|r| \approx 0.8$–$1.0$), as reported in Table 5, substantially outperforming training loss or its moving average. This confirms that routing dynamics remain tightly coupled with downstream generalization.

The nanoMoE results reinforce our central claim: *post-memorization routing reorganization is not an artifact of model scale, data size, or specific MoE implementation.* Instead, it reflects an inherent property of MoE training dynamics. Combined with the large-scale OLMoE study, these

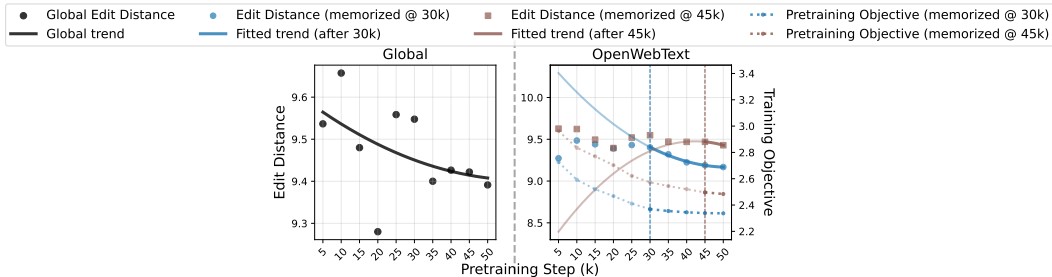

Figure 14: **Pathway edit distance and pretraining objective on nanoMoE during pretraining.** The left panel reports the overall pathway edit-distance trajectory across checkpoints without grouping samples by their memorization steps. The right panel instead visualizes pathway dynamics for two memorized data groups (memorized at 30k and 45k steps). After their respective memorization points, both groups exhibit a consistent post-memorization decline in edit distance despite a plateaued pretraining objective, indicating that nanoMoE continues to reorganize routing pathways toward more structured and shared computation. This mirrors the memorization-to-generalization transition observed in large-scale OLMoE.

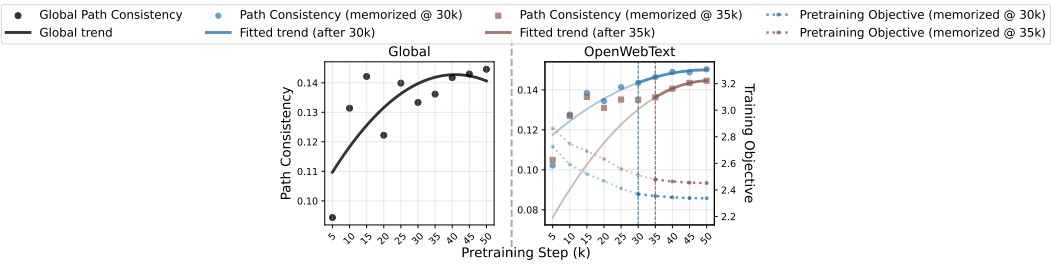

Figure 15: **Evolution of pathway consistency in nanoMoE.** The left panel reports the overall progression of pathway consistency across checkpoints. The right panel further analyzes two memorized data groups (memorized at 30k and 35k steps), showing that once memorization is achieved, each group undergoes continued refinement of its layer-to-layer routing transitions.

findings establish pathway metrics as stable, architecture-intrinsic indicators of the memorization-to-generalization transition.

Table 5: **Correlation Between Pathway Metrics and Downstream Accuracy on nanoMoE** across four commonsense benchmarks (ARC-Easy, ARC-Challenge, HellaSwag, OpenBookQA). $p$-value: $p < 0.1$ , $p < 0.05$ , $p < 0.01$ , $p > 0.1$ , $p > 0.5$ , $p > 0.8$ . Preferred correlation: positive , negative .

| Metric | ARC-Easy | | ARC-Challenge | | HellaSwag | | OpenBookQA | |
|---|---|---|---|---|---|---|---|---|
| | Pearson | Spearman | Pearson | Spearman | Pearson | Spearman | Pearson | Spearman |
| Pathway Distance (pairwise) | -0.8113 | -1.0000 | -1.0000 | -1.0000 | -0.9713 | -1.0000 | -0.8366 | -0.9000 |
| Pathway Consistency (sample-wise) | 0.9416 | 1.0000 | 0.8818 | 1.0000 | 0.9234 | 0.8000 | 0.9979 | 1.0000 |
| Training Loss | 0.3301 | -0.1000 | -0.7799 | -0.5000 | -0.2182 | -0.2000 | 0.5984 | 0.8000 |
| Training Loss (moving avg.) | 0.4664 | 0.4000 | -0.6078 | -0.8000 | -0.3580 | 0.2000 | -0.8655 | -0.4000 |

# LLM USAGE STATEMENT

In preparing this manuscript, we used LLM solely as a writing assistance tool. Specifically, the model was employed to aid in grammar correction, sentence refinement, and improving clarity of exposition. It did not contribute to research ideation, experimental design, data analysis, or interpretation of results.

