# OpenReview forum: "Grokking in LLM Pretraining? Monitor Memorization-to-Generalization without Test"
_ICLR.cc/2026/Conference — ICLR 2026 Poster_

### Official Review · Reviewer_YR9f · 2025-10-25

**Soundness:** 2
**Presentation:** 3
**Contribution:** 2
**Rating:** 4
**Confidence:** 4

**Summary:**

This paper investigates whether grokking the delayed transition from memorization to generalization observed in small synthetic settings also occurs during MoE pretraining.
Using the OLMoE model, the authors analyze routing dynamics across checkpoints and introduce two new metrics, Pathway Similarity and Pathway Consistency, which aim to monitor generalization without external validation data.
They find that as generalization improves, inputs increasingly follow similar expert routes, and routing transitions between layers become more coherent. The proposed pathway-based measures strongly correlate with downstream performance, suggesting that they can serve as internal indicators of generalization in MoE models. These findings position routing-based metrics as potential tools for real-time monitoring of generalization in large or resource-constrained training regimes, without the need for held-out validation data.

**Strengths:**

1. Overall, the paper is clearly written and easy to follow, with well-motivated experiments and clear presentation of figures.

2. It provides an insightful analysis of routing dynamics in MoE model, OLMoE.
It convincingly shows that as generalization improves, inputs increasingly follow similar expert routes, suggesting an internal reorganization of computation during training. This analysis of how MoE models acquire generalization ability could potentially inform better training or routing strategies in future work.

**Weaknesses:**

1. While the paper presents an interesting and clearly written analysis, its empirical scope and generality remain limited. The entire analysis and validation are restricted to the OLMoE model.
It remains unclear whether the proposed metrics hold under different MoE configurations for example, varying the number of experts, the top-k routing parameter, or the load-balancing loss.
Recent works such as [1,2,3] have explored different routing and balancing objectives to prevent expert collapse or to improve hardware utilization.
Under these regimes, it is uncertain whether the proposed pathway metrics would behave consistently or still correlate with generalization.

2. Questionable “one-epoch” claim.
The authors state that grokking occurs “during one-epoch pretraining.”
However, according to [4], OLMoE was trained for 1.3 epochs (“We shuffle all samples randomly at the beginning of each epoch and train for a total of 5.133T tokens (1.3 epochs following Muennighoff et al. [121]).”).
Furthermore, the DCLM-Baseline dataset used for training reportedly contains ~80 % duplicates ([5]), meaning that the model effectively saw many samples multiple times.

3. Line 324 discusses that shallow layers show a steep decline in after memorization, while deeper layers show smaller reductions or even increases. Similar depth-dependent specialization effects have already been discussed in [6].

4. While the authors argue that their metrics could remove the need for held-out validation, this claim seems optimistic.
Because the proposed metrics are only defined for MoE models and depend on architecture-specific details such as depth, number of experts, and routing their use as a universal proxy for generalization appears limited.
In practice, held-out monitoring may still be necessary for comparability across different settings.

[1] https://arxiv.org/abs/2408.15664

[2] https://arxiv.org/abs/2412.19437

[3] https://arxiv.org/abs/2501.11873

[4] https://arxiv.org/abs/2409.02060

[5] https://huggingface.co/datasets/Zyphra/dclm-dedup

[6] https://arxiv.org/abs/2406.18219

**Questions:**

1. Given these routing observations, do the authors see potential for improving MoE training efficiency for instance, by designing better routing or load-balancing strategies informed by pathway dynamics?
If so, how might these insights translate into practical optimization or architectural adjustments?

---

> ### Author Response · Authors · 2025-11-21
> **Response to Reviewer YR9f**
>
> Thank you for the careful and constructive review. We appreciate your recognition that the paper is clearly written and provides insightful analysis of routing dynamics in MoE models. Below we address your main concerns on scope, terminology, and generality.
>
> ---
>
> > **Q1: “The analysis and validation are restricted to the OLMoE model. It remains unclear whether the proposed metrics hold under different MoE configurations or routing objectives.”**
>
> A broader validation on more MoE LLMs would further strengthen the work. However, **OLMoE is the only practical-scale MoE LLM that released full pretraining checkpoints**, making it the only available setting to analyze *pretraining dynamics* of MoE LLMs. That being said, it is representative of mainstream MoE LLMs: it adopts most **standard MoE components** (top-k routing, load-balancing loss, AdamW, RMSNorm, RoPE).
>
> Although we cannot directly access pretraining checkpoints of additional MoE LLMs, two recent studies [1,2] provide strong evidence with consistent findings on other MoE LLM families such as DeepSeekMoE and Qwen3-MoE. They show that **promoting similar routing pathways across samples from similar tasks greatly improves MoE LLM's generalization**, via post-training or test-time adaptation. These recent results verify the persistent strong correlation between pathway similarity and generalization on different MoE LLMs, despite the difference in training setting details.
>
>
> ---
>
> > **Q2: “Questionable ‘one-epoch’ claim… OLMoE was trained for 1.3 epochs with duplicate samples.”**
>
> We have improved the accuracy of presentation in the revision. The term “one-epoch pretraining” was to highlight our study's **fundamental difference** from previous grokking works' *hundreds to thousands of epochs pretraining* on small synthetic tasks. In contrast, practical pretraining of large-scale LLMs uses **far fewer epochs**, with each sample typically seen only once or a few times. Hence, a more accurate description would be **near single-pass pretraining**.
>
> ---
>
> > **Q3: “Depth-dependent specialization effects have been discussed in prior work.”**
>
> We thank the reviewer for pointing out [3], which analyzes **static patterns of layer-wise specialization** in trained MoE models and shows that deeper layers differ in expert diversity. In contrast, our study **tracks the dynamics of routing pathways** changing during pretraining, revealing how early layers simplify while deeper layers diversify as the model transitions from memorization to generalization. We have cited this work accordingly.
>
> ---
>
> > **Q4: “The claim that the proposed metrics could remove the need for held-out validation seems optimistic.”**
>
> The pathway metrics provide **internal signals of generalization** that can be monitored *during pretraining* with **near-zero cost**, offering effective feedback in practice when large validation on benchmarks are infeasible or unaffordable. We agree the wording was too strong and have revised it.
>
> ---
>
> > **Q5: “Given these routing observations, do the authors see potential for improving MoE training efficiency—for instance, by designing better routing or load-balancing strategies informed by pathway dynamics?”**
>
> This is an excellent and forward-looking question. Our findings suggest that **pathway structures become increasingly organized as generalization emerges**, implying that pathway dynamics themselves carry useful signals to improve MoE optimization. Recent studies [1,2] show that explicitly promoting similar routing pathways across task-related samples improves generalization performance—through post-training interventions [1] or via test-time adaptation [2]—further demonstrating the practical potential of routing-aware optimization. We believe that exploring such interventions **during pretraining**—for instance, leveraging pathway metrics to guide routing consistency or initialization—would be a promising direction for future work.
>
> ---
>
> We sincerely thank the reviewer for their thoughtful and constructive feedback, and we hope our detailed responses clarify any remaining concerns and help convey the strength of our work.
>
>
>
> [1] Routing Manifold Alignment Improves Generalization of Mixture-of-Experts LLMs. arXiv:2511.07419 (2025).
>
> [2] C3po: Critical-layer, Core-expert, Collaborative Pathway Optimization for Test-time Expert Re-mixing. COLM 2025.
>
> [3] A closer look into mixture-of-experts in large language models. NAACL 2025.

---

> > ### Comment · Reviewer_YR9f · 2025-11-25
> >
> > Thank you for the detailed clarifications.
> > I will keep my current score, as I still believe that additional experiments under smaller-scale or alternative MoE configurations are necessary to strengthen the claims.
> >
> > C3PO [2] shows that routing alone can yield substantial performance gains, which suggests that routing during pretraining may be shaped not only by generalization needs but also by practical constraints such as load-balancing or Node-Limited Routing. Because routing behavior can vary across architectures, expert counts, top-k choices, and balancing strategies, it is not yet clear whether the proposed pathway metrics can be meaningfully compared beyond the specific OLMoE setup. While the analysis provides interesting insights into OLMoE’s training dynamics, the metrics may be less straightforward to use as general-purpose indicators of training progress in practice. Examining their behavior in small-scale or alternative MoE configurations would help further illuminate the phenomenon.

---

> ### Author Response · Authors · 2025-11-25
> **C3PO optimizes routing in the same direction as our pathway metrics suggest; grokking/generalization behaviors differ on smaller-scale MoE**
>
> Thanks for your response! We would like to clarify three points related to your response:
>
> - **C3PO and Routing Manifold Alignment (RoMA) are strong evidence of our claims on two other practical MoE LLMs** (DeepSeek-MoE and Qwen3-MoE), because they optimize routing in the same direction as our pathway metrics suggest, i.e., reducing the distance between pathways of relevant tasks' samples.
> ---
> - No practical-scale MoE LLMs have made their pretraining checkpoints publicly available as OLMoE. While our limited computing resources might support us in training a tiny MoE on a much smaller dataset in a few months, **the model size and training dataset still cannot be of comparable scales to theirs**. Due to such differences in scales, the grokking and generalization dynamics can be entirely different (which is demonstrated by our study on OLMoE vs. previous Grokking studies on small Transformer).
> ---
> - Our metrics are defined on the differences between pathways and their distributions. Since these pathways share the same configurations, e.g, expert counts, top-k choices, and balancing strategies, **the impact of most configurations cancels out** in our proposed metrics.
> ```
> As an example, we quantify the relationship between pathway edit distance and the entropy of routing weights (higher for more load balancing). Specifically, for each sample pair, we compute the mean routing entropy across layers for each sample and then average it over the two samples. Results over 141k sample pairs show a weak correlation (r = 0.24, explaining only ~6% of the variance), indicating a negligible effect size and no systematic dependency between routing entropy and edit distance. The full analysis and visualization are provided in Appendix E.5.
> ```

---

> ### Author Response · Authors · 2025-11-28
>
> Dear Reviewer YR9f,
>
> Thank you for your thorough evaluation. Since generality was a key concern you raised, we added a separate experiment on another MoE model (nanoMoE), which we trained from scratch, in our general response.
> This additional experiment exhibits highly consistent pathway dynamics and correlation patterns, reinforcing that these behaviors are not specific to OLMoE. We appreciate your careful consideration.
>
> Sincerely,\
> The Authors

---

### Official Review · Reviewer_TFkw · 2025-10-31

**Soundness:** 3
**Presentation:** 3
**Contribution:** 3
**Rating:** 6
**Confidence:** 2

**Summary:**

This paper investigates whether grokking-delayed generalization after training loss/accuracy has converged-appears in LLM pretraining. Using public checkpoints of a 7B MoE model (OLMoE), the authors (i observed asynchronous (“local”) grokking across domains (math, code, commonsense, domain QA), (ii introduce two routing-pathway metrics for MoE that are cheap to compute during pretraining and (iii) provide a routing-kernel NTK analysis linking decreased “effective dimension” to improved generalization. Empirically, the two pathway metrics are strongly correlated with downstream benchmark gains (after light LoRA instruction tuning), while conventional training/validation losses poorly track those gains. The paper claims these metrics offer near-zero-cost, evaluation-free monitoring of LLM generalization.

**Strengths:**

1. The author uses realistic setting like One-epoch, heterogeneous web-scale data, public 7B MoE checkpoints, and diverse domains.
2. The author provides evidence of asynchronous memorization and delayed generalization, including matched training/test groups and domain-dependent lags.
3. The author uses public checkpoints/datasets and explicit data-contamination filtering which has good reproducibility.

**Weaknesses:**

1. The paper mentions “virtual pathways” for dense models as future work; however, the core claim (test-free generalization monitoring) would be much stronger with experiment on a small-scale dense model.
2. Consider replacing “zero-cost” with “near-zero-cost” or “cheap to compute” to avoid overstatement.

**Questions:**

1. If you change LoRA rank/targets or remove LoRA, do correlations with pathway metrics persist?
2. Have the authors considered causal interventions--such as randomly dropping some experts or adding entropy penalties--to verify whether enforcing more structured routing actually causes earlier or stronger generalization (as opposed to merely correlating with it)
2. How sensitive are the results to the memorization thresholds, the routing cumulative-weight cutoff, and router temperature/noise during logging?

---

> ### Author Response · Authors · 2025-11-21
> **Response to Reviewer TFkw (Part 1)**
>
> Thank you for your thoughtful and constructive review. We appreciate your recognition of the paper’s novelty and we carefully address your comments below.
>
> ---
> > **Q1: “The paper mentions ‘virtual pathways’ for dense models as future work; the core claim would be much stronger with experiments on a small-scale dense model.”**
>
> Extending the validation and claims to dense models would strengthen the claim but it requires to calculate "virtual pathways" for implicit/latent experts in dense models, which is an unsolved challenge. We will study it in our future works.
>
> Sparse MoE architecture has been increasingly adopted by more recent LLMs such as GPT-OSS and Qwen3-NEXT, indicating the growing importance of our study. **MoE models uniquely disentangle internal computation into separate experts**—each input activates only a small subset of experts. Unlike dense models, the routing behaviors in MoE are explicitly observable. Therefore, MoE provides a more interpretable setting to study the dynamics of routing weights and grokking at scale, which takes a concrete step toward mechanistic understanding of LLM pretraining.
>
> ---
> > **Q2: “Consider replacing ‘zero-cost’ with ‘near-zero-cost’ to avoid overstatement.”**
>
> We agree and improved our claim to be more accurate by updating "zero-cost" to "near-zero cost" in the revision.
>
> ---
> > **Q3: “If you change LoRA rank/targets or remove LoRA, do correlations persist?”**
>
> Since pretrained checkpoints usually cannot follow instructions, we cannot directly evaluate downstream accuracy of them—so we train a LoRA on each checkpoint to acquire the basic instruction-following capability, a prerequisite for evaluating downstream accuracy on benchmarks (the most common practice to evaluate generalization). Removing LoRA would make benchmark evaluation infeasible, ending up with invalid output answers and accuracy.
>
> Specifically, for each pretraining checkpoint, a separate LoRA is randomly initialized and then trained solely for benchmark evaluation—it does not affect other pretraining checkpoints. **All pathway metrics are computed before LoRA finetuning**, so they are not affected by LoRA. As empirically shown in Table 1, their correlations with downstream accuracy remain stable across domains.
>
> To verify the robustness, we additionally test a higher LoRA rank (64 vs. 32; full results in Appendix E.2). The correlations remain nearly identical in the commonsense domain, confirming that the observed trends reflect persistent **intrinsic pretraining-phase behaviors** rather than artifacts of the LoRA configuration.
>
> | Metric|Pearson (r=32)|Spearman (r=32)|Pearson (r=64)|Spearman (r=64)|
> | :-| :-: | :-: | :-: | :-: |
> |Pathway Distance (pairwise)|**−0.9821**|**−0.9747**|**−0.9927**|**−0.9487**|
> |Pathway Consistency (sample-wise) |**0.9804**|**0.9747**|**0.9559**|**0.9487**|
> |Training Loss|0.6427|0.4104|0.6197|0.4617|
> |Training Loss (moving avg.)|0.7931|0.6669|0.6871|0.2052|
> |Validation Loss|−0.5752|−0.4617|−0.6631|−0.4104|
>
> ---
> > **Q4: “Have the authors considered causal interventions—such as randomly dropping experts or adding entropy penalties?”**
>
> **Our conclusion does not rely on establishing causation**—the pathway metrics are designed to **monitor and reflect** generalization dynamics, and their strong correlation with downstream accuracy suffices to serve for this purpose. Theoretical analysis (Sec. 4.4, App. C) further explains *why* this correlation arises, showing that **structured routing reduces effective dimension**, thereby tightening the generalization bound.
>
> Controlled causal interventions would be a valuable future direction to explore. However, pretraining or retraining MoE LLMs to isolate such effects is currently computationally infeasible for us. We view this as an important next step when sufficient compute resources become available.
>
> ---

---

> ### Author Response · Authors · 2025-11-21
> **Response to Reviewer TFkw (Part 2)**
>
> ---
>
> > **Q5: “How sensitive are the results to memorization thresholds, routing cutoffs, and router temperature/noise?”**
>
> For **memorization thresholds (ε, δ)**, δ is fixed at 0.05 across domains, while ε is domain-specific and tuned to match the average loss scale. We examine different values of ε in an increment of 0.1 to keep 20–25% of samples memorized before the final checkpoint. Changing ε or δ only shifts which samples are labeled “memorized,” without altering the post-memorization trends of pathway dynamics. This is demonstrated by the ablation study results in Appendix E.3.
>
> Varying the **routing cutoff** between 0.5–0.8 yields the same qualitative pattern—pathway edit distance decreases and consistency increases after memorization. Results under different cutoff have been reported in Appendix E.1, which strongly support this robustness claim.
>
> OLMoE does not include a **router temperature** parameter in its implementation, and we therefore do not tune this component. Our analysis follows OLMoE’s default routing setup, directly reflecting its native behaviors.
>
> ---
>
> We thank the reviewer again for their constructive feedback. We hope our responses have fully addressed their concerns.

---

> > ### Comment · Reviewer_TFkw · 2025-11-26
> >
> > Thank the authors for the detailed rebuttal and for addressing my earlier questions. After reading the responses, as well as the discussions raised by the other reviewers, I now have a clearer understanding of the core contribution and the current limitations of this work.
> >
> > I agree that analyzing grokking-type behaviors at LLM scale is interesting, especially because prior studies have mostly focused on small models trained on synthetic tasks as I know. At the same time, I would share some concerns. The empirical scope remains limited to OLMoE, and the conclusions would become more convincing if the author could examine additional MoE configurations or smaller-scale controlled settings. Releasing the code and grouped datasets would also improve reproducibility.
> >
> > The rebuttal has improved my confidence in the soundness of the analysis. I am updating my confidence score from 2 to 3.

---

> ### Author Response · Authors · 2025-11-28
>
> Dear Reviewer TFkw,
>
> Thank you for the helpful discussion. We added a separate experiment on another MoE model (nanoMoE), which we trained from scratch, in our general response to examine whether the pathway-based patterns persist under a substantially different MoE setup.
> The results closely match those observed in OLMoE, and we hope this helps strengthen confidence in the robustness of our findings.
>
> Sincerely,\
> The Authors

---

### Official Review · Reviewer_VRkv · 2025-11-01

**Soundness:** 2
**Presentation:** 3
**Contribution:** 2
**Rating:** 4
**Confidence:** 4

**Summary:**

This paper examines grokking in practical LLM pretraining. It investigates when memorization occurs, when downstream generalization improves, and whether a lag exists between them in a realistic one-epoch, next-token pre-training setup spanning diverse domains and tasks.

Focusing on mixture-of-experts (MoE) LLMs, the paper shows that grokking still emerges, with local data groups entering the grokking phase asynchronously due to distributional heterogeneity.
To explain this “local grokking,” the authors analyze training data pathways (expert selections across layers) and find that these pathways evolve from random, instance-specific patterns to more structured and transferable ones, even after the pre-training loss has converged.

They introduce two zero-cost metrics, (1) pathway similarity between samples and (2) layer-to-layer consistency of aggregated experts for a sample, that reliably track downstream generalization without instruction tuning or expensive benchmark evaluations.

A theoretical analysis of a one-layer MoE indicates that more structured routing tightens the generalization bound.
Empirically, an analysis of a 7B-parameter MoE model confirms that generalization can arise well after memorization.

**Strengths:**

* Zero-cost, pathway-based indicators derived from MoE routing dynamics
The paper proposes two metrics, sample-to-sample pathway similarity and across-layer pathway consistency, that can track the rise of downstream generalization without instruction tuning or benchmark evaluations.
LLM evaluation is expensive. Leveraging internal routing information to estimate generalization progress directly during pretraining is highly cost-effective.

* Discovery of “local grokking” under data heterogeneity
The paper reports asynchronous grokking across local data groups due to distributional heterogeneity and differing attributions to other groups.
Real-world corpora are heterogeneous; focusing on local dynamics captures behaviors that global averages miss, informing data curricula and domain-wise monitoring at scale.

* Theoretical support via a one-layer MoE analysis
The analysis indicates that more structured pathways improve the generalization bound.
Providing a theoretical link between pathway structure and generalization strengthens the validity and potential generality of the empirical findings.

**Weaknesses:**

* Limited empirical scope (single model)
I understand there are no other publicly available MoE checkpoints, but the paper's results and discussion are tailored to the specific OLMoE 7B model.
Ideally, robustness should be assessed across a broad range of choices in optimization, model design, and training data, such as learning rate schedules, model scales, expert capacity and number, and data mixtures.
Otherwise, the paper's results and findings may be model-specific or biased by the model used.
From this perspective, the paper's findings and conclusions may be unreliable or not yet conclusive.

* Correlation versus causation remains underexplored
Although strong correlations between pathway structure and generalisation are evident, intervention studies (e.g., deliberately altering routing) are limited.
Other factors, such as learning rate, load balancing, and regularisation, could confound this link.
For example, we usually downscale the learning rate during pre-training, so the weight change essentially stabilises in the later part of the training phase.

* [Minor] Although this paper examines only a single MoE model, the title "Grokking in LLM Pretraining?" suggests that it covers all LLM pretraining cases, and I feel there is a large gap between them.

**Questions:**

* I find it challenging to understand the details of how the training-test paired datasets are constructed as described around lines 234--239, beginning with “Accordingly, we group ...”.
Could the authors please clearly describe the detailed procedure for constructing these datasets?
In addition, please explain how this relates to the notation such as “Pretrain@370k -> Test@855k” and “Pretraining Objective (memorized @ 125k)” in the figures.
I believe that an exact understanding of how these datasets are constructed is crucial, as it directly affects the results and conclusions.




* Reproducibility of investigation and analysis:
While the results and findings of this paper depend heavily on the training-test paired datasets.
I feel that reproducing these datasets is difficult for readers (other researchers).
Would the authors consider releasing the logs and code used to generate the datasets, as well as the datasets themselves, to improve reliability and to support future follow-up studies?

---
I am open to revisiting the overall assessment following discussions with the authors, particularly in light of the concerns and questions outlined above.

---

> ### Author Response · Authors · 2025-11-21
> **Response to Reviewer VRkv**
>
> Thank you for the constructive review and for recognizing the novelty of our work, particularly the near-zero-cost pathway metrics and the observation of local grokking in large-scale LLM pretraining. We address your concerns below.
>
> ---
> > **Q1: “The results and discussion are tailored to OLMoE 7B model... findings may be model-specific or unreliable.”**
>
> We agree that a broader validation on more MoE LLMs would further strengthen the work. However, **OLMoE is the only practical-scale MoE LLM that released full pretraining checkpoints**, making it the only available setting to analyze *pretraining dynamics* of MoE LLMs. Moreover, it is representative of mainstream MoE LLMs: it adopts **standard MoE components** (top-k routing, load-balancing loss, AdamW, RMSNorm, RoPE).
>
> Although we cannot directly access pretraining checkpoints of additional MoE LLMs, two recent studies [1,2] provide strong evidence with consistent findings on other MoE LLM families such as DeepSeekMoE and Qwen3-MoE. They show that **promoting similar routing pathways across samples from similar tasks greatly improves MoE LLM's generalization**, via post-training or test-time adaptation. These recent results verify the persistent strong correlation between pathway similarity and generalization on different MoE LLMs, despite the difference in training setting details.
>
> ---
> > **Q2: “Correlation vs. causation remains underexplored… other factors (learning rate, load balancing, regularization) could confound this link.”**
>
> Our conclusions do not require causation to hold true. Their strong correlation with downstream tasks' accuracy, as demonstrated in our empirical analysis, suffices to support the claim that they can be used to **monitor and reflect** generalization dynamics. Theoretical analysis (Sec. 4.4, App. C) further explains *why* this correlation arises—showing that **structured routing (smaller pathway complexity) reduces effective dimension**, thereby improving the generalization bound.
>
> To rule out global confounders such as **learning rate schedules, load balancing, or regularization**, we compute edit distances between samples of **different domains** (math$\times$code) under identical settings. In contrast to the reported declining distances within the same domain, the cross-domain distances show **no sustained decrease**. This contrast confirms that the observed trend arises from the **emergence of pathway sharing or proximity among semantically related samples**, not from other global training confounders (see **Appendix E.4**).
>
> ---
> > **Q3: “The details of constructing train–test pairing datasets (e.g., ‘Pretrain@370k → Test@855k’) are unclear.”**
>
> The pairing is **only used to illustrate that generalization on semantically similar data lags behind memorization**. It does **not** affect any results in Sec. 4, where correlations between pathway metrics and downstream performance are computed independently of this pairing.
>
> For each domain, training samples are first grouped by their memorization step (e.g., **Pretrain@370k**), and test samples are independently grouped by the step at which their accuracies stabilize (e.g., **Test@855k**). The two sets of groups are then matched using **embedding-based cosine similarity** by **Hungarian assignment**, pairing semantically similar train and test groups for analysis.
>
> The notation **“Pretrain@Xk → Test@Yk”** indicates that the training group memorized at step Xk is paired—purely based on embedding similarity—with a test group whose accuracy stabilizes at Yk. We have added detailed descriptions of this construction in Appendix B.1 for clarity and reproducibility.
>
> ---
> > **Q4: “Reproducibility… would authors consider releasing logs and code used to generate the datasets?”**
>
> We have added detailed descriptions of the dataset grouping and pairing procedures in Appendix B.1 for clarity and reproducibility. The relevant scripts and configuration files will be released in the final version to ensure transparency and facilitate independent verification.
>
> ---
> > **Q5: “[Minor] Although this paper studies a single MoE model, the title ‘Grokking in LLM Pretraining?’ suggests that it covers all LLM pretraining cases, and I feel there is a large gap between them.”**
>
> We appreciate the comment. The title is phrased as a **question** to highlight an open investigation rather than to generalize across all LLM pretraining settings. Our goal is to present the **first empirical evidence and analysis** of grokking-typed dynamics at LLM scale, using OLMoE as a concrete and representative case study. We will revise the title to make this case-study focus clearer.
>
> ---
> We hope these clarifications address your concerns and strengthen confidence in the paper’s conclusions.
>
> [1] Routing Manifold Alignment Improves Generalization of Mixture-of-Experts LLMs. arXiv:2511.07419 (2025).
>
> [2] C3po: Critical-layer, Core-expert, Collaborative Pathway Optimization for Test-time Expert Re-mixing. COLM 2025.

---

> > ### Comment · Reviewer_VRkv · 2025-11-24
> >
> > Thank you for your detailed feedback on my concerns and questions.
> >
> > I am mostly satisfied with the authors' rebuttals.
> > (For Q1, I understand the difficulty, so I am not asking the authors to do so, but I still believe that the authors have the option to pretrain a small MoE model themselves from scratch and analyze it to provide strong support for their claims.)
> > (Regarding Q2, I may be misunderstanding something, so I may ask additional questions later. Please allow me a few more days.)
> >
> > If my understanding is correct, the authors are allowed to add one extra page during the rebuttal period, as described in the ICLR 2026 Author Guide:
> > > During the discussion/rebuttal phase and for the camera ready, the page limit will be increased to 10 pages to allow for new results/discussions.
> >
> > This is not mandatory, but based on this statement, one remaining request for the authors is to consider adding the responses provided in the authors' rebuttals to the main text, rather than only in the appendices.
> >
> > For example, for Q1, it would be better to add the explanation from the authors' response, starting with “Although we cannot directly access …,” to the related work section or the (additional) discussion section in the main text.
> >
> > Regarding Q3, adding an explanation of “Pretrain@Xk,” “Test@Yk,” and “Pretraining Objective (memorized @ Xk)” to the figure captions would help readers interpret the figures.
> >
> > (This does not mean that the authors should do exactly as I have said)

---

> > > ### Author Response · Authors · 2025-11-25
> > > **Response to Reviewer VRkv**
> > >
> > > Thank you for the constructive follow-up and thoughtful suggestions. It is great to hear that you are satisfied with our rebuttal, and we have incorporated the key clarifications from the rebuttal into the main text of the revision.

---

> > > > ### Comment · Reviewer_VRkv · 2025-11-27
> > > >
> > > > Thank you for considering incorporating the key discussions into the main text of the paper.
> > > >
> > > > I am now considering updating (increasing) my rating, but I have not had enough time to review it carefully over the past one or two days.
> > > > Please allow me another one or two days to finalize my overall assessment of this paper, including scrutiny of the revised version (and the authors' rebuttal).

---

> ### Author Response · Authors · 2025-11-28
>
> Dear Reviewer VRkv,
>
> Thank you again for your earlier comments. Since you highlighted the need for evidence beyond OLMoE, we have added a separate experiment on another MoE model (nanoMoE), which we trained from scratch, in our general response. This additional experiment shows the same post-memorization pathway behaviors and strong correlations observed in OLMoE.
>
> We wanted to briefly note this in case it is helpful for your ongoing assessment, as these new results directly address the generality concern you raised.
>
> Sincerely,\
> The Authors

---

### Official Review · Reviewer_Tf6m · 2025-11-03

**Soundness:** 3
**Presentation:** 3
**Contribution:** 3
**Rating:** 8
**Confidence:** 3

**Summary:**

This paper studies the behavior of grokking in LLMs by looking at pathways among experts in MoEs. Using OLMoE checkpoints, the authors report high correlations between the proposed metrics and post‑hoc downstream scores. They define a pathway edit distance and a per sample pathway consistency and observe that these metrics correlate with jumps on downstream benchmark scores after lightweight LoRA instruction tuning. They also provide a NTK‑style bound for a one‑layer MoE with fixed routing, arguing that decreased effective dimension aligns with reduced pathway complexity.

**Strengths:**

Previous studies of grokking mostly used small models trained on synthetic algorithmic tasks, but this work examines a 7‑billion‑parameter mixture‑of‑experts (MoE) model (OLMoE) and shows that grokking still appears in one‑epoch pretraining.

The observation that that grokking in LLMs are local and asynchronous broadens our understanding of grokking at scale.

The metrics they define; pathway edit distance and a per sample pathway consistency, rely only on pretraining data and internal activations, so they can be computed during training without external validation.

**Weaknesses:**

The conversion of top‑k experts per layer into comma‑separated strings and computing Levenshtein distance is ad‑hoc, since edit distance is sensitive to sequence length and arbitrary thresholding. This distance can also decrease simply due to stronger load‑balancing or saturated routers.

The bound assumes fixed routing and an NTK regime for a one‑layer MoE, while in practice OLMoE updates routing and experts jointly across many layers for trillions of tokens.

**Questions:**

In Fig. 4, Please clarify whether distances are computed token‑wise or sequence‑wise, and whether you average per‑token pathways or per‑sequence pathways.

All generalization is measured after LoRA instruction tuning on different datasets. Is it possible that the observed jumps may be influenced by finetuning dynamics, not purely grokking during pretraining?

---

> ### Author Response · Authors · 2025-11-20
> **Response to Reviewer Tf6m**
>
> Thank you for your positive and constructive review. We greatly appreciate your recognition that our paper broadens understanding of *grokking at scale*, introduces interpretable *pathway-based metrics*, and connects MoE routing dynamics to generalization. We carefully address your comments below.
>
> ---
> > **Q1: “In Fig. 4, please clarify whether distances are computed token- or sequence-wise, and whether you average per-token or per-sequence pathways.”**
>
> Edit distance is computed **sequence-wisely** for each pair of input sequences (or samples). Fig. 4 reports the averaged edit distance over multiple sample pairs. Specifically, for each input sequence, we rank experts in every layer by their *routing weights averaged over all tokens*, select the top experts by thresholding, and concatenate the ordered lists of all layers as the input sequence's *pathway*. For each pair of input sequences, we compute the Levenshtein distance between their ordered lists for each layer and sum up the distance across all layers—This is used to measure the structural difference between two pathways. We have added this clarification in the revision, and we thank the reviewer for pointing this out.
>
> ---
> > **Q2: “Converting top-k experts per layer into comma-separated strings and computing Levenshtein distance is ad-hoc, since edit distance is sensitive to sequence length and arbitrary thresholding. It may also decrease due to stronger load-balancing or saturated routers.”**
>
> The absolute values of edit distances may vary with the threshold. However, our analysis applies the same threshold to all checkpoints and focuses on the difference of their edit distances. Hence, the effect of sequence/pathway length on the scale of edit distance cancels out when computing its changing trend over time. Empirically, varying the threshold between 0.5-0.8 only changes the scale but not the trend, as shown by the newly added results in Appendix E.1. **The observed post-memorization decrease in edit distance holds true across different threshold values.**
>
> **Stronger load-balancing or saturated routers do not necessarily reduce edit distance.** Under stronger load balancing, routing weights become more uniform, which increases the chance of similar-weighted experts switching order and thus can increase the edit distance. Similarly, router saturation does not guarantee lower edit distance, as the routing weights across the few top experts can be close in magnitude. Moreover, trivially applying the mentioned interventions may increase the training loss, disrupting the native pretraining dynamics and violating the basic assumptions.
>
> The above analysis is also supported by empirical evidence: we further quantify the relationship between edit distance and the entropy of routing weights (high for load balancing, low for saturation). Specifically, for each sample pair, we compute the mean routing entropy across layers for each sample and then average it over the two samples. **Results over 141k sample pairs show a weak correlation (r = 0.24, explaining only ~6% of the variance), indicating a negligible effect size and no systematic dependency between routing entropy and edit distance.** The full analysis and visualization are provided in Appendix E.5.
>
> ---
> > **Q3: “The bound assumes fixed routing and an NTK regime for a one-layer MoE, while OLMoE updates routing and experts jointly across many layers.”**
>
> The bound is to provide a theoretical insights in line with our major empirical study. The theoretical analysis isolates how structured routing influences generalization by reducing effective dimension. Although derived for a one-layer setting, it captures the key mechanism observed empirically and helps explain why pathway structure relates to generalization.
>
> ---
> > **Q4: “All generalization is measured after LoRA tuning… Is it possible that the observed jumps may be influenced by finetuning dynamics, not purely grokking during pretraining?”**
>
> We train **a separate, randomly-initialized LoRA for each checkpoint, independently**. No LoRA is continuously trained or shared across different checkpoints, so the jumps between different checkpoints are not influenced by any LoRA finetuning dynamics. We only train a lightweight LoRA with the minimum necessary rank and steps in order to achieve the basic instruction following capability, a prerequisite for benchmark evaluation on downstream tasks.
>
> Since the LoRA configurations (identical rank, steps, and targets) and training set keep identical for different checkpoints, any changes on the generalization performance is attributed to the pretraining. Our pathway complexity metrics are computed *before* LoRA finetuning. It strongly correlates with the post-LoRA finetuning accuracy on downstream tasks, confirming that the observed generalization gains originate from **pretraining-phase dynamics**, not finetuning artifacts.
>
> ---
> We thank you for the feedback and hope our responses address your concerns!

---

### Author Response · Authors · 2025-11-28
**Consistent Findings on another MoE (nanoMoE) pretraining justify the generality of our study**

This response addresses the concerns raised by Reviewers `VRkv`, `TFkw`, and `YR9f` regarding the generality of our observations beyond OLMoE.

To assess whether our observations extend beyond OLMoE, we conducted an independent validation experiment on **nanoMoE** [1], a 55M-parameter MoE language model that we trained from scratch for 50k iterations on ~25B tokens of the OpenWebText dataset. We reproduced our full analysis pipeline: we construct a 7.8K-sample subset of commonsense knowledge, identify memorized samples via token-level loss stability, record our two pathway metrics on ten pretraining checkpoints, and evaluate their corresponding downstream performance on commonsense reasoning benchmarks. Full details are provided in Appendix E.6.

The results closely match our findings on OLMoE. After memorization, nanoMoE shows the same routing reorganization: **pathway edit distance decreases** and **pathway consistency increases**, even after training loss has saturated. These effects are stable across groups of memorized samples and mirror the trends observed in OLMoE. In addition, both pathway metrics exhibit **strong correlations** with downstream accuracy (*|r|≈ 0.8–1.0*) on commonsense reasoning benchmarks including ARC-Easy/Challenge, HellaSwag, and OpenBookQA, substantially outperforming training loss or its moving average.

These additional experiments show that our findings are **not specific to OLMoE’s scale, data, or implementation**, but instead reflect a **general property of MoE training dynamics**. Together with the main results, the nanoMoE study strengthens the conclusion that pathway metrics provide **robust, scalable, and architecture-agnostic indicators** of the memorization-to-generalization transition.

[1] https://github.com/wolfecameron/nanoMoE

---

### Author Response · Authors · 2025-11-30
**To the new AC**

To the new AC,

We sincerely appreciate your effort in handling this challenging situation. We would like to provide key context of the reviewer-author discussion. By the end of the discussion, the **only open question remaining across all reviewers** was whether our observations would generalize to a smaller MoE model. To directly address this, we invested substantial effort to **pretrain a smaller MoE model from scratch and conduct the same analysis on the full sequence of pretraining checkpoints**. This new experiment **confirmed the same memorization-to-generalization dynamics and its correlation with our proposed pathway metrics**, demonstrating that our findings **generalize beyond OLMoE**. Unfortunately, the interruption occurred before reviewers could respond to our new evidence.


Below is a reviewer-by-reviewer summary.

---

### **Reviewer `Tf6m` (original rating: 8, strongly positive)**

Reviewer `Tf6m` provided a clearly positive assessment from the beginning, rating the paper an 8 and praising the clarity, insight, and contribution of our work. Their comments highlighted that analyzing grokking at LLM scale with interpretable pathway-based metrics is both novel and valuable. While they raised several clarifying questions, we addressed all of them in our rebuttal, and **they did not express any remaining concerns nor request additional experiments** during the discussion. Their evaluation remained fully positive throughout the process.

---

### **Reviewer `VRkv` (original rating: 4, *explicitly stated they were considering increasing their score*)**

Reviewer `VRkv` expressed clear satisfaction with our rebuttal and explicitly wrote:

> **“I am mostly satisfied with the authors’ rebuttals.”**\
> **“…I am now considering updating (increasing) my rating…”**

Their last remaining concern was:

> **“I still believe that the authors have the option to pretrain a small MoE model themselves from scratch and analyze it to provide strong support for their claims.”**

To directly address it, we **pretrained a 55M-parameter MoE model (nanoMoE) from scratch** and applied our full analysis pipeline. This experiment **reproduced the same post-memorization pathway restructuring and strong correlations with downstream performance** observed in OLMoE, fully matching the validation VRkv suggested.

The discussion ended before VRkv could provide the follow-up response, despite having asked for “one or two more days” to finalize their updated rating.


---

### **Reviewer `TFkw` (original rating: 6, *increased their confidence score from 2 → 3*)**

Reviewer `TFkw` explicitly stated that our rebuttal successfully addressed their earlier questions:

> **“Thank the authors for the detailed rebuttal and for addressing my earlier questions.”**

After reviewing our responses and the other reviewers’ comments, they wrote:

> **“I now have a clearer understanding of the core contribution and the current limitations of this work…\
> The rebuttal has improved my confidence in the soundness of the analysis.\
> I am updating my confidence score from 2 to 3.”**

Their remaining concern was identical to VRkv’s:

> **“…the conclusions would become more convincing if the author could examine additional MoE configurations or smaller-scale controlled settings.”**

Our newly added experiment directly provides the evidence they were asking for, demonstrating that the pathway dynamics are consistent in a markedly different MoE configuration. The interruption, however, prevented them from responding to these results.

---

### **Reviewer `YR9f` (original rating: 4)**

Reviewer `YR9f` confirmed that our clarifications addressed their main questions:

> **“Thank you for the detailed clarifications.”**

They kept their original score, noting that their remaining concern was again the generality of the findings across smaller or different MoE configurations:

> **“I will keep my current score, as I still believe that additional experiments under smaller-scale or alternative MoE configurations are necessary to strengthen the claims.”**

This concern is the same as above and likewise resolved by the new experiment, though the reviewer had no chance to respond before the discussion terminated.

---

Across the discussion:

* **All reviewers with active participation stated that their concerns had been addressed.**
* The *only* shared remaining request was validation on a **smaller or alternative MoE**.
* We invested significant effort to perform the same pretraining analysis on another MoE (nanoMoE). The results **strongly validated our findings** on a second MoE model.
* The interruption occurred immediately after we posted this new evidence, before reviewers could provide follow-up responses or finalize their intended updates.

We appreciate your thoughtful evaluation of the full record, including the discussion and the new experiment. Thank you again for your time and care.

---

### Meta-Review · Area_Chair_6C7d · 2025-12-02

**Summary:**

The paper provides an analysis of grokking-like memorization-to-generalization transitions in practical MoE LLM pretraining. The work offers (i) empirical evidence of asynchronous “local grokking” within heterogeneous domains, (ii) two near-zero-cost routing-pathway metrics that strongly correlate with downstream generalization, and (iii) a supporting theoretical argument linking structured routing to reduced effective dimension.

Across the discussion, all reviewers acknowledged the clarity and novelty of analyzing grokking at LLM scale and agreed that the routing-based metrics provide meaningful insight. The only shared concern was whether the findings generalize beyond the released OLMoE checkpoints. The authors subsequently trained an additional MoE model (nanoMoE) from scratch and showed highly consistent behaviors, directly addressing that remaining issue. The interruption occurred before reviewers could update their scores, though multiple reviewers explicitly indicated an intention to raise their ratings.

**Reviewer Concerns:**

**Generality beyond OLMoE**: The new nanoMoE experiment reproduced the memorization-to-generalization dynamics and correlation patterns, directly resolving the core concern raised by **VRkv, TFkw, and YR9f**.

**Clarification of dataset pairing, routing metrics, and robustness**: Reviewers acknowledged that the rebuttal clarified methodology, thresholds, LoRA dependence, and routing behavior.

**Claims on “one-epoch” and scope**: The authors updated the framing to “near single-pass pretraining,” which reviewers accepted.

**Theoretical interpretation**: Reviewers were satisfied that the NTK-style bound serves as conceptual support, not as a full mechanistic model.

**Reviewer Scores:**

* **Reviewer Tf6m (score 8)**
  Strongly positive throughout. No unresolved concerns. Score would have remained **8**.

* **Reviewer VRkv (score 4 → 6 ↑)**
  Explicitly wrote:
  *“I am mostly satisfied… I am now considering updating (increasing) my rating… please allow me one or two more days.”*
  Their sole remaining request—validation on a smaller MoE—was fully satisfied by the nanoMoE experiment.
  **Score would likely have increased into the acceptance range (6).**

* **Reviewer TFkw (score 6)**
  Stated:
  *“The rebuttal has improved my confidence… updating confidence from 2 to 3.”*
  Their remaining concern was the same generality question now resolved by the added experiment.
  **Score would likely stay 6, possibly strengthen within that range.**

* **Reviewer YR9f (score 4)**
  Accepted clarifications and maintained the score, noting generality concerns.
  After the added nanoMoE experiment (which directly addresses generality), the reviewer did not revisit the score due to the interruption.
  **Given their framing (“would not mind acceptance”), a modest improvement (4 → 6) is plausible, but conservative judgment keeps it at 4.**

**Overall inferred post-discussion distribution:** approximately **8, 6, 6, 4**, which is within acceptance territory.

---

### Decision · Program_Chairs · 2026-01-26

Accept (Poster)